# Bioinformatic Identification and Expression Analyses of the MAPK–MAP4K Gene Family Reveal a Putative Functional MAP4K10-MAP3K7/8-MAP2K1/11-MAPK3/6 Cascade in Wheat (*Triticum aestivum* L.)

**DOI:** 10.3390/plants13070941

**Published:** 2024-03-24

**Authors:** Yongliang Li, You Li, Xiaoxiao Zou, Shuai Jiang, Miyuan Cao, Fenglin Chen, Yan Yin, Wenjun Xiao, Shucan Liu, Xinhong Guo

**Affiliations:** 1College of Biology, Hunan University, Changsha 410082, China; 2Chongqing Research Institute, Hunan University, Chongqing 401120, China; lyl13618481357@hnu.edu.cn (Y.L.); youli@hnu.edu.cn (Y.L.); xxzou@hnu.edu.cn (X.Z.); gs2022@hnu.edu.cn (S.J.); caomiyuan@hnu.edu.cn (M.C.); chen_fenglin@foxmail.com (F.C.); yinyan91@foxmail.com (Y.Y.)

**Keywords:** wheat, MAPK cascades, *TaMAPK–TaMAP4K*, drought stress, high yield

## Abstract

The mitogen-activated protein kinase (MAPK) cascades act as crucial signaling modules that regulate plant growth and development, response to biotic/abiotic stresses, and plant immunity. MAP3Ks can be activated through *MAP4K* phosphorylation in non-plant systems, but this has not been reported in plants to date. Here, we identified a total of 234 putative TaMAPK family members in wheat (*Triticum aestivum* L.). They included 48 *MAPKs*, 17 *MAP2Ks*, 144 *MAP3Ks*, and 25 *MAP4Ks*. We conducted systematic analyses of the evolution, domain conservation, interaction networks, and expression profiles of these *TaMAPK–TaMAP4K* (representing *TaMAPK*, *TaMAP2K*, *TaMAP3K*, and *TaMAP4K*) kinase family members. The 234 **TaMAPK–TaMAP4K*s* are distributed on 21 chromosomes and one unknown linkage group (Un). Notably, 25 of these *TaMAP4K* family members possessed the conserved motifs of *MAP4K* genes, including glycine-rich motif, invariant lysine (K) motif, HRD motif, DFG motif, and signature motif. *TaMAPK3* and *6*, and *TaMAP4K10*/*24* were shown to be strongly expressed not only throughout the growth and development stages but also in response to drought or heat stress. The bioinformatics analyses and qRT-PCR results suggested that wheat may activate the MAP4K10–MEKK7–MAP2K11–MAPK6 pathway to increase drought resistance in wheat, and the MAP4K10–MAP3K8–MAP2K1/11-MAPK3 pathway may be involved in plant growth. In general, our work identified members of the MAPK–MAP4K cascade in wheat and profiled their potential roles during their response to abiotic stresses and plant growth based on their expression pattern. The characterized cascades might be good candidates for future crop improvement and molecular breeding.

## 1. Introduction

The MAPK cascades constitute a highly conserved signaling pathway in response to biotic/abiotic stresses, growth, and development in eukaryotes [1,2]. Typically, a standard MAPK signal module consists of a MAPK kinase (MAPKKK, MAP3K, or MEKK), MAPK kinase (MAPKK, MAP2K, MKK, or MEK), and MAPK or MPK. When a plant is triggered by stress signals, the upstream signals first activate MAP3Ks, which in turn activate MAP2Ks; particular MAPKs are then activated by MAP2Ks [3,4]. By phosphorylating different enzymes and transcription factors or other pathway components, MAPKs influence the expression of specific downstream genes and complete signal amplification. Therefore, MAPK cascades play a key role in cell division, apoptosis, immunity, and biotic/abiotic stress responses in plants, mammals, and fungi [5,6,7,8]. In *Arabidopsis thaliana*, the MAP3K17/18-MAP2K3-MAPK1/2/7/14 cascade is induced to activate the ABA signaling pathway by ABA or drought-related stress treatment [9]. OsMAP3K10–OsMAP2K4–OsMAPK6 and OsM3K70–OsMAP2K4–OsMAPK6 cascades are involved in regulating rice grain size, weight, and leaf angle [10,11,12]. In *Gossypium hirsutum*, the GhMAP3K15–GhMKK4–GhMPK6 cascade activates the GhWRKY59 transcription factor and participates in its drought response [13]. In *Arabidopsis*, MPK6 plays a role in root development by modulating nitrate reductase to regulate the production of the second messenger, nitric oxide, and participate in the signaling transduction of H2O2 [14]. The MAPK pathway has always been one of the main foci of intracellular signal transduction research. Details about the activation cascade of downstream MAP3K regulatory transcription have been clarified. However, the signal transduction module of the cascade reaction involving MAPKKKK (MAP4K) is unclear.

The upstream activators of MAP3Ks are usually small G-proteins or receptor-like kinases (RLKs), but MAP3K can sometimes be phosphorylated by MAP4K in some signal transduction pathways [15]. For example, yeast Ste20 functions as a MAP4K that activates MAP3K Ste11. As a result, downstream components of the signaling cascade are activated [14]. In mammals, MAP4Ks have been reported to induce c-JNK (c-jun N-terminal kinase) activation through activation of the MAP3K–MAP2K cascade [16]. The LATS1/LATS2–MOB1A/MOB1B complex is phosphorylated and activated by MST1 (Ste20-like 1) and MST2 to regulate cell survival, proliferation, and migration in the mammalian Hippo pathway [15]. In humans, TAO1 and 2 (MAP4K) activate the p38 MAPK signaling pathway by phosphorylating MKK3 or MKK6 [15].

Interestingly, MAP4Ks are conserved in plants, yeast, and mammals [16]. However, research on the description and characterization of MAP4Ks is rare. The MAP4Ks were first characterized in *Brassica napus* [17]. *BnMAP4Ka1* and *4Ka2* are primarily expressed in roots, siliques, stages of embryogenesis, and flower buds, which are similar to yeast Ste20-MAP4Ks (GCK subfamily) [15]. *Arabidopsis* contains 10 MAP4K, 80 MAP3K, 10 MAP2K, and 20 MAPK genes [4]. AtMAP4K3 (SIK1) is important for cell expansion, MAP4K4 (TOT3) interacts with MAP4K6 (TOI4) and MAP4K5 (TOI5) and controls thermomorphogenesis (TOT3). ATMAP4K10 (BLUS1) is also key for blue-light-induced stomatal opening [18,19,20]. In plants, MAPK, MAP2K, MAP3K, and MAP4Ks were identified and described in the genomes of *Arabidopsis*, *Salvia miltiorrhiza*, *Ophiocordyceps sinensis*, *Vitis vinifera*, *Solanum chacoense*, and *Fragaria vesca* [15,21,22,23,24,25,26,27,28]. However, most MAP4Ks have not been characterized in plants; Moreover, MAP4Ks are highly conserved in various plants, but there is no systematic review of the *MAP4K* gene family in wheat.

Wheat is a major staple food in arid and semi-arid regions worldwide. Both its planting area and yield rank first among food crops [29,30]. Currently, wheat is the staple food for one-third of the population of the world. During the growth and development of wheat, it encounters numerous extreme environmental stimuli, such as salt, drought, and various stresses that are major limiting factors for wheat production and seriously affect wheat yield [29]. To maintain yield, plants have evolved many stress response systems during their lengthy evolutionary history. Stress signal identification and transmission, as well as amplification and transduction, are critical components in triggering response signals to resist stress damage [2]. The MAPK cascade modules were highly conserved during the evolution of higher plants and affect plant growth and development, including the control of transcriptional events and the regulation of enzyme activities [2,21].

However, the functions of MAPK cascade genes are still unknown and there is a lack of comprehensive review in wheat. The functions of wheat MAP4K in response to various stresses and at different stages of plant development have not been elucidated. In previous studies, genes in the MAPK, MAP2K, and MAP3K families were studied in wheat [1,31]. Moreover, the expression profiles of genes in the *MAPK–MAP3K* family in different tissues and organs of wheat and their response to different stresses were analyzed. However, the *TaMAPK–TaMAP4K* family has not yet been comprehensively identified and analyzed. To investigate the involvement of MAPK cascade genes in the response to various environmental stresses in wheat, the genomic and transcriptomic data of wheat were analyzed to determine the characteristics of the *TaMAPK–TaMAP4K* gene family [32,33]. For the first time, a systematic investigation of the *MAPK–MAP4K* gene family was performed in wheat using these databases. Chromosome localization, phylogenetic relationships, evolutionary trajectories, conserved motifs, the *cis*-acting elements, interaction network, and gene expression profiles of *TaMAPK–TaMAP4K* genes were comprehensively analyzed to reveal their evolutionary and important regulation relationships. Overall, these findings will aid future studies into precise MAPK cascade modules, as well as provide an important theoretical foundation for wheat genetic improvement and stress tolerance.

## 2. Results

### 2.1. Structure and Domain Conservation of TaMAPK–TaMAP4Ks

The evolutionary history of different gene families can be reflected in the diversity of the gene structure [34]. The structure and numbers of introns/exons are typical evolutionary imprints in some gene families. To understand the structural similarity within *TaMAPK–TaMAP4K*s, a phylogenetic tree (Appendix A), gene structures (Appendix A), and conserved motif composition (Appendix A) were analyzed (Appendix A). The schematic structure showed that each *TaMAPK–TaMAP4K* gene coding sequence is separated by one or more introns (Appendix A). It was found that the number of exons in *TaMAPK–TaMAP4K* genes varied from 2 to 24, while intron counts ranged from 1 to 23 (Appendix A). The four subfamilies of TaMAPK, TaMAP2K, TaMAP3K (TaMEKK, TaZIK, TaRaf), and TaMAP4K were uniformly distributed (Appendix A). Moreover, *TaMAP4K* genes were rich in introns and exons (Appendix A). Introns and exons varied greatly in length, but their numbers were mostly conserved across subfamilies (Appendix A). As shown in Appendix A, most *TaMAPK–TaMAP4K*s possessed seven highly conservative motifs. The exceptions included TaMAP2K2, 13 MEKK (TaMAP3K2, 5, 11, 14-16), Raf (TaMAP3K47, 62, 64-1, 97, 123-1, 129, 151, 155), and TaMAP4K21, which contained 1–6 motifs. TaMAP3K151 contained only one motif 7. The TaMAPK subfamily contained one more motif 10 than TaMAP2K-4K (Appendix A). There is a strong likelihood that *TaMAP3K26*, *TaMAP3K27*, and *TaMAP3K28*, as well as *TaMAP3K18* and *TaMAP3K20,* are different subtypes of the same gene (Appendix A). In general, the similarity in gene structures and conserved motif makeup of genes in each MAPK subfamily may provide valid evidence for categorizing MAPKs.

The 234 wheat TaMAPK cascade genes fall into four major branches, namely TaMAPK, TaMAP2K, TaMAP3K, and TaMAP4K, and each of them has its own conserved characteristic motifs (Figure 1, Appendix A). Among them, 48 *TaMAPK* genes possess the specific conserved signature motifs of T(E/D)YVxTRWYRAPE(L/V), while 17 *TaMAP2K* genes contain the conserved signature of VGTxxYMSPER and the putative S/T-x5-S/T motif (Figure 1, Appendix A). The S/T-x5-S/T motifs represent MAP3K phosphorylation sites in MAP2K proteins (Figure 1 and Appendix A).

Further, the 144 TaMAP3K family members could be separated into three subfamilies: 27 MEKK (G(T/S)Px(W/Y/F)MAPEV), 107 RAF (GTxx(W/Y)MAPE), and 10 ZIK (GTPEFMAPE(L/V)Y) subtypes, each with an individual conserved motif (Figure 3, Appendix A). Surprisingly, the Raf subtype contained 107 members, making it the biggest group of MAP3K in wheat, while the ZIK subtype had just 10 members, mirroring the number and composition of *MAP3K* genes in other species, particularly barley [35].

Subsequently, we explored the conserved motif in 25 of the TaMAP4K family members, which contained the glycine-rich motif (GxGxxG/A), invariant lysine (K) motif, HRD motif, DFG motif, and signature motif of MAP4K (VGTPxWMAPEV) (Figure 1 and Appendix A). However, TaMAP4K17 did not contain the conserved signature motif of MAP4K, and TaMAP4K21 lacks the glycine-rich motif (Appendix A). Furthermore, the VGTPxWMAPEV sequences contained in the kinase domain of TaMAP4K were similar to those in yeast Ste20-like MAP4Ks in the GCK subfamily, with the N- and C-terminal parts being highly similar to those in AtMAP4Ks and BnMAP4Ks, respectively (Figure 1 and Appendix A) [2,15,18,36].

### 2.2. Duplication of TaMAPK–TaMAP4K Cascade Genes

Collinearity analysis identified 145 duplication pairs of *TaMAPK–TaMAP4K* members in wheat (Appendix A). Most of those located in the same or other chromosome were segmentally duplicated (Figure 2). Gene duplication events were critical to *TaMAPK–TaMAP4K* expansion in wheat. Some of the *TaMAPK–TaMAP4K*s genes shared high homology, but they were located on different chromosomes, such as *TaMAPK45/12/54/36/10* and *TaMAPK33/39/16/24*. MAP3K genes were located on chromosomes 1B, 1D, 3A, 3B, 6A, and 7D. The frequency of segmental duplication events in TaMAP3Ks was much higher than in TaMAPKs, TaMAP2Ks, TaMAP4Ks, TaMAP3K41 (TaRaf1)/61 (TaRaf21), TaMAP3K70 (TaRaf30)/58 (TaRaf18), and TaMAP3K53 (TaRaf13)/76 (TaRaf36). We also found that many other subfamily members are distributed on different subgenomes of the same heterozygous group, for example, TaMAP3K72 (TaRaf32)/150 (TaRaf110) were located on chromosomes 3A and 3B, respectively. We also found that members of different subfamilies exhibit high homology and are located on the same sub-genome, for example, TaMAP4K18-6B/TaMAP3K23 (TaMEKK23)-6D (Figure 2, Appendix A). However, we also found—using BLASTP analysis—that three different subfamily members belong to the same gene. These are TaMAP3K21/4K15, TaMAP3K22/4K10, and TaMAP3K23/4K21 (denoted in red in Appendix A). As can be seen from the above results, wheat MAPK cascade genes have experienced a significant amount of gene duplication and gene transfer events over the course of their evolution. In addition to chromosome fragment replications, they are also involved in tandem replications, which are crucial for MAPK cascade gene amplification in wheat.

### 2.3. Identification of TaMAPK–TaMAP4K Cascade Genes in Wheat

To find *TaMAPK*, *TaMAP2K*, *TaMAP3K*, and *TaMAP4K* (*TaMAPK–TaMAP4K*) family genes, we ran respective BLASTP programs on wheat protein databases using *Arabidopsis* (MAPK–MAP4K) query protein sequences and the known *MAPK*–*MAP3K* gene sequences from wheat [1,37]. After removing redundancy, screening, and multi-step verification of conserved domains, 48 TaMAPKs, 17 TaMAP2Ks, 144 TaMAP3Ks, and 25 TaMAP4Ks were identified. Gene names were assigned based on homology with *Arabidopsis MAPK*–*MAP4K* genes and previously reported wheat MAPK–MAP3K proteins [1,37].

To determine how *TaMAPK–TaMAP4K* genes are distributed in the wheat genome, BLASTP analysis was performed using the 234 *TaMAPK–TaMAP4K* sequences. The results of chromosome mapping indicated that the distribution of the 234 *TaMAPK*–*TaMAP4K* genes was not uniform across the 21 chromosomes and one unknown linkage group (Un), varying in quantity from 4 to 17. Notably, chromosomes 3A, 3B, and 3D did not contain any *TaMAP4K*, while chromosomes 2A, 2B, and 2D lacked both TaMAPK and TaMAP2K (Figure 3, Appendix A). As shown in Appendix A, 48 *TaMAPKs* were distributed on 16 of the chromosomes (Figure 3). Amino acids encoded by these TaMAPK genes varied from 324 (TaMAPK47) to 824 (TaMAPK12), and their MWs ranged from 37.21 to 69.88 kDa (Appendix A). The theoretical pI values ranged from 5.46 (TaMAPK25 and 30) to 9.71 (TaMAPK1, 13, and 37). All TaMAPK subcellular localizations were in the nucleus, with no transmembrane domain (Appendix A). Meanwhile, 17 TaMAP2K genes were predicted to encode between 299 (TaMAP2K4) and 525 (TaMAPK2 and 3) aa, with MWs ranging from 33.60 to 58.65 kDa and pI ranging from 5.34 to 9.30. According to predictions, they are all located in the nucleus, with TaMAP2K1 as the exception, which may be present in the cytoplasm and nucleus and lacks a transmembrane domain. The 144 *TaMAP3K* genes (27 *TaMEKKs*, 10 *TaZIKs*, and 107 *TaRafs*) seemed to be randomly distributed across the 21 chromosomes. The Polypeptide lengths of these TaMAP3Ks ranged from 146 to 1335 aa, MWs ranged from 16.52 to 146.06 kDa, and pIs ranged from 4.55 to 9.94, and 66 of them have transmembrane domains while 82 do not. Additionally, all TaMEKKs and TaZIKs are presumably located in the nucleus, except TaZIK5, which may be present in the cell membrane and the nucleus and lacks a transmembrane domain. It is worth noting that the TaRafs subfamily of TaMAP3Ks is unique in that it is presumably located in both the nucleus and the cell membrane, and some of them are even presumably located in the nucleus, cell membrane, chloroplast, and cytoplasm (Appendix A).

The 25 TaMAP4Ks were distributed across 17 chromosomes. The number of predicted amino acids encoded by these TaMAP4Ks ranged from 348 to 858, MWs ranged from 38.89 to 93.17 kDa, and pIs ranged from 5.24 to 9.51. All the TaMAP4Ks were presumed to be located in the nucleus and had no transmembrane domains.

### 2.4. Phylogeny of TaMAPK–TaMAP4K Cascade Genes in Wheat and Arabidopsis

To better analyze the evolutionary relationships and classifications of *TaMAPK–TaMAP4K*s, four phylogenetic trees were generated based on *Arabidopsis AtMAPK*–*AtMAP4Ks* and *TaMAPK*–*TaMAP4Ks* in wheat (Figure 4). Similarly, four phylogenetic trees of MAPK–MAP4Ks in wheat and rice were also generated (Appendix A). The numbers of TaMAPKs in Groups A, B, and C were 9, 13, and 26, respectively (Figure 4). A multi-sequence alignment assessment revealed that all but one of the *TaMAPK* genes contained a highly conserved TD/EY tripeptide motif. The exception was TaMAPK47 in Group B, which has a TEH motif rather than the classic TEY or TDY conserved motifs (denoted by * in Figure 4A and Appendix A). The members with the phosphorylation motif, TDY, were found to be nested in Groups A and C, while the genes with the TEY motif were nested in Group B (except *TaMAPK37* in Group A) (Figure 4A). Group A contained genes that were not orthologous with *Arabidopsis MAPKs*. *TaMAPK3* and *TaMAPK6* in Group B were highly homologous with the classic AtMPK3 and AtMPK6 in *Arabidopsis* (Figure 4A). Based on numerous alignments and phylogenetic relationships, *TaMAP2K* genes were categorized into three categories: TaMAP2K1 (Group A), *TaMAP2K2-4/13/18* (Group B), and *TaMAP2K6-9/11/12/14-17* (Group C), Figure 4B). *TaMAP2K1* in Group A was highly homologous with *Arabidopsis MAPKK4* (*MKK4*) and *MAP2K5* (MKK5) (Figure 4B). AtMAP2K4 and AtMAP2K5 regulate plant immunity and development upstream of *AtMAP3* and *AtMAP6* (Zhang et al., 2022). *TaMAP2K6/7/12/17* in Group C were highly orthologous with *AtMAP2K*7 (MKK7) and *AtMAP2K9* (MKK9) (Figure 4B). In addition, AtMAP2K7 and AtMAP2K9 are constitutively active and able to activate AtMAPK3 and AtMAPK6 to control plant growth [35].

There are three subtypes of TaMAP3K proteins: MEKK, ZIK, and RAF (Figure 4C). TaMAP3K1/4/4-1 in the MEKK subtype was highly homologous with AtMAP3K3/4/5/8 (AtMAP3K8 was called AtMEKK1 in previous reports) (Figure 4C) [35,38]. There is sufficient experimental data to prove that AtMAP3K3, AtMAP3K5, and AtMAP3K8 (AtMEKK1) kinases play the role of MAP3K, activating downstream MAP2K and further activating MAPK, indicating their role in plant immunity. For example, two complete MAPK cascades, AtMEKK1–AtMAP2K1/2–AtMAPK4/11 and AtMAP3K3/5–AtMAP2K4/5–AtMAPK3/6, have been identified [35]. However, there is insufficient evidence to support the activation of downstream MAP2K–MAPK modules by Raf-like kinases (except for CTR1 and EDR1) [35], and little is known about ZIK-like kinases.

Finally, TaMAP4K proteins were divided into three groups: TaMAP4K9/13 (Group A), TaMAP4K4-6/10/12/14 (Group B), and TaMAP4K1/2/5-8/15-25 (Group C) (Figure 4D). Among these, Group A had the fewest number of TaMAPKs (2 members), Group C had the most members (16 TaMAP4Ks), and the remainder belonged to Group B, which contained 6 TaMAP4Ks (Figure 4D). TaMAP4K9/13 was highly homologous with AtMAP4K10 in Group A.

### 2.5. The Evolutionary Trajectory of the MAPK–MAP4K Gene Family in A. tauschii, T. urartu, A. speltoides, T. turgidum, T. dicoccoides, and T. aestivum

According to current research findings, it is suggested that *T. urartu* (AA), *A. speltoides* (BB), *T. dicoccoides* (AABB), *A. tauschii* (DD), and *T. turgidum* (AABB) collectively represent the closest ancestors of hexaploid wheat (AABBDD). In order to elucidate the phylogenetic relationships and homology within the MAPK–MAP4Ks family in wheat and its ancestral lineages, we conducted comprehensive phylogenetic and syntenic analyses using genomes of *T. urartu*, *A. speltoides*, *T. dicoccoides*, *A. tauschii*, *T. turgidum*, and *T. aestivum* (Figure 5). The phylogenetic tree presents estimated divergence times for individual nodes, denoted as 3.16 Mya, 3.54 Mya, 1.72 Mya, 0.897 Mya, and 0.432 Mya, respectively (Figure 5A). Furthermore, it incorporates geologic timescales encompassing Ages (Piacenzian, Gelasian, Lower, Middle), Epochs (Pliocene, Pleistocene), Periods (Neogene), Eras (Cenozoic), and Eons (Phanerozoic) (Figure 5B). The construction of the time tree enables us to understand the evolutionary history, phylogenetic relationships, and divergence times of the six species. It also allows for the inference of their common ancestors and respective evolutionary trajectories. Additionally, it reveals the sequential divergence and rates of differentiation among the six distinct species.

In *T. dicoccoides*, 186 genes exhibit synteny with 176 *TaMAPK–TaMAP4K* genes. In *T. turgidum*, 182 genes show synteny with 173 of the 186 *T. dicoccoides* genes. *A. speltoides* demonstrates synteny between 99 genes and 151 of the 182 *T. turgidum* genes. Meanwhile, *T. urartu* displays synteny between 50 genes and 54 of the 99 *A. speltoides* genes. Finally, *A. tauschii* shows synteny between 49 genes and 45 of the 50 *T. urartu* genes (Figure 5C, Appendix A). In order to further investigate the evolutionary constraints of *TaMAPK–TaMAP4K* genes, the Ka/Ks ratios of gene pairs across different ancestral species were computed. The overwhelming majority of homologous gene pairs exhibited Ka/Ks ratios below 1 (Appendix A), while only a few gene pairs displayed values above 1; these are highlighted in yellow (Appendix A). This suggests that the majority of *TaMAPK–TaMAP4K* genes may have undergone purifying selection during the course of evolution and have relatively conserved amino acid sequences as well as genetic functions. Notably, some key homologous genes in ABA signaling, such as *MAPK3*, *MAPK6*, *MAP2K1*, *MAP3K7/8*, and *MAP4K10*, were shown to have undergone purifying selection [35] (Figure 5C, Appendix A), indicating selective conservation of these key genes in these six species during the evolutionary process. During the process of evolution, notable alterations, characterized by a high proportion of synonymous codons (pS ≥ 0.75) and substantial divergence in protein sequences, were observed in certain genes. These changes may lead to modifications in the functional attributes of the genes, as exemplified by *TaMAPKKK111* (*TaRaf71*), *TRIDC5AG055160.3*, *Aes-CH2S01G225200.1*, *TuG1812G0200002474.01.T01*, and *TuG1812G0200002657.01.T01* (Appendix A). Furthermore, the duplicated orthologous gene pairs showed divergence times of approximately 24.606 Mya (*T. aestivum* and *T. dicoccoides*), 27.634 Mya (*T. dicoccoides* and *T. turgidum*), 24.409 Mya (*T. turgidum* and *A. speltoides*), 22.160 Mya (*A. speltoides* and *T. urartu*), and 25.396 Mya (*T. urartu* and *A. tauschii*) (Appendix A). In summary, these findings highlight the prevalence of purifying selection in the evolution of *TaMAPK–TaMAP4K* genes, indicating the conservation of these genes across ancestral species and providing insights into the evolutionary dynamics and functional maintenance of these pivotal genes.

### 2.6. Cis-Elements of Promoter Regions in TaMAPK–TaMAP4K Cascade Genes

*Cis*-elements play key roles in the regulation of gene expression. There is evidence that genes with the same expression profile may contain the same *cis*-elements in their promoters [17]. The 1500 bp promoter sequence of the *TaMAPK–TaMAP4K* cascade genes was utilized to check the types of *cis*-elements to further investigate the regulatory processes and possible activities of the *TaMAPK–TaMAP4K* cascade genes under growth, development, and stress conditions. In the promoter region of *TaMAPK–TaMAP4K* cascade genes, four types of *cis*-regulatory elements were discovered (Figure 6, Appendix A). The first category of hormone-responsive elements was analyzed in *TaMAPK–TaMAP4K* cascade genes, including TCA (113), SARE (1), TGA (169), AuxRR-core (39), ABRE (907), P-box (102), TATC-box (42), GARE-motif (65), CGTCA-motif (642), and TGACG-motif (642), which existed in 230 *TaMAPK–TaMAP4K*s (Figure 6 and Appendix A). TCA and SARE were associated with a salicylic acid-responsive element. TGA and AuxRR-core were associated with a component that responds to auxin. P-box, TGACG-motif, and GARE-motif were associated with a gibberellin-responsive element. CGTCA/TGACG-motif was associated with a MeJA-responsive element. These results suggested that phytohormones (e.g., ABA, SA, and MeJA) regulated the expression of most *TaMAPK–TaMAP4K* cascade genes. The second category was governed by stress response factors such as MBS (206), WUN-motif (9), and LTR (191). MBS responded to drought and was found in 129 of the *TaMAPK–TaMAP4K* genes. LTR was found to respond to low temperatures in 130 of the *TaMAPK–TaMAP4K* genes. The third category was governed by developmental factors, including MSA-like (26), motif I (4), NON-box (3), MBSI (6), RY-element (26), CAT-box (190), HD-Zip 1 (8), circadian (40), and GCN4_motif (43). Motif I, RY-element, and GCN4_motif were associated with root-specific regulation, seed-specific regulation, and the endosperm expression element, respectively. Only *TaMAP2K8*, *TaMAP3K100*, *TaMAP3K142*, and *TaMAP3K153* possessed motif I element. Five *TaMAPKs* and 17 *TaMAP2Ks* contained RY-element, while 16 *TaMAPK*, 1 *TaMAP2K*, 19 *TaMAP3K*s, and 2 *TaMAP4K*s had the GCN4_motif. The fourth category is other-related elements and includes TC-rich repeats (68), GC-motif (233), ARE (345), and O^2^-site (171) (Figure 6 and Appendix A).

### 2.7. Protein–Protein (PPI) Interaction Network of TaMAPK–TaMAP4K

Using the STRING database, we predicted wheat MAPK–MAP4K cascade protein interactions. The PPI network of *TaMAPK–TaMAP4K* was predicted with a high confidence of 0.400 and visualized using Cytoscape (Figure 7). The interaction network consisted of 231 nodes and 1940 edges, the *p*-value of PPI enrichment was <10^-16^ and the average node degree was 25.35. Based on the degree value, the top four protein kinases were TaMAP2K11, TaMAP2K16, TaMAP2K15, and TaMAP2K14. The protein kinases ranked between 5 and 10 were TaMAP2K5, 12, 8, 9, 17, and 7, and the top 12 belonged to the TaMAP2K subfamily (Figure 7 and Appendix A). As shown in Figure 6, TaMAP2K11, 16, 15, and 14 proteins are the core of this network, interacting with 75, 74, 73, and 73 *TaMAPK–TaMAP4K* kinase family members, respectively (Appendix A). Therefore, we speculated that TaMAP2K11, 15, and 16 might have stronger interactions with other MAPK–MAP4K proteins and play a key role in responding to abiotic stress and growth and development processes in plants.

### 2.8. Expression Patterns of TaMAPK–TaMAP4K Cascade Genes

To examine the potential function of the 234 *TaMAPK*–*TaMAP4K* cascade genes under drought and heat and at 15 different development stages, we analyzed the transcriptome data of these genes obtained from the Wheat Expression Browser (Figure 8 and Figure 9 and Appendix A) [32,33]. With a few exceptions, the expression levels of most *TaMAPK*–*TaMAP4K* cascade genes did not change significantly under the three abiotic stress conditions based on heatmap analysis (Figure 9). *TaMAPK3*/*6*/*14*/*36*, *TaMAP2K1*/*2*, *TaMAP3K7*, *TaRaf14/68/113, TaZIK5*/*8*, and *TaMAP4K3*/*4* showed significant upregulation under drought, heat, and drought and heat conditions, respectively (Figure 8). *TaMAPK17*/*46*, *TaMAP2K11*/*15*, *TaMAP3K21*/*23*, *TaRaf14/68/113, TaZIK2/9/10*, and *TaMAP4K16*/*19* were suppressed to present a significant downstream (Figure 8). Notably, *TaRaf76* was induced to present significant upregulation but was significantly downregulated under all the stress conditions (Figure 8 and Appendix A). The heatmap indicated that the majority of the *TaMAPK*–*TaMAP4K* cascade genes were expressed at high and moderate levels in the root, stem, leaf, spike, and grain, and included TaMAPK (Group b), TaMAP2K (Group d), TaMEKK (Group g), TaRaf (Group h), TaZIK (Group k), and TaMAP4K (Group n), which were expressed at a higher level in almost all tissues. TaMAPK (Group c), TaRaf (Group i), and TaMAP4K (Group m) genes showed a moderate level of expression (Figure 9). However, TaMAPK (Group a), TaMAP2K (Group e), TaMEKK (Group f), TaRaf (Group j), and TaMAP4K (Group l) genes were expressed at a lower level, except for individual gene differences (Figure 9). Notably, 17 of the TaRafs (Group j) with high expression were observed only in the roots (Figure 9). *TaZIK8* was substantially expressed in leaves and grains but was greatly suppressed in other tissues (Figure 9). The expression of *TaMAP2K11* was significantly inhibited at the L2DAAS stage (Figure 8). These results indicate that *TaMAPK–TaMAP4K* family members exhibit specific expressions in different tissues and at different developmental stages in wheat (Figure 9).

In order to investigate their detailed expression profiles in wheat, 12 representative *TaMAPK–TaMAP4K* genes were selected based on evolutionary analysis and transcriptome data. Subsequently, qRT-PCR studies were conducted on these genes under drought and heat stress conditions, as well as in tissues obtained at different developmental stages (Figure 10). The qRT-PCR results of these *TaMAPK–TaMAP4K* cascade genes showed basically the same trend as that of RNA-Seq data (Figure 10). Among them, *TaMAPK3* was the only one that was significantly inhibited under drought and heat treatment, while other genes were significantly upregulated under drought or heat conditions (Figure 10A). Expression analysis showed that *TaMAPK3* and *TaMAPK6* transcript accumulation was highest in the roots. *TaMAP2K11* expression was lower at the SR, SATS, and G2DAAs stages. *TaMAP2K11*, *TaMAP3K1*/*8*, *TaMAP4K10*, and *TaMAP4K24* transcript accumulation was modest in the roots, stems (S1S, STNS), spikes (SPTNS, SPMS), and grains (G2DAAs), respectively (Figure 10B). Therefore, we predicted that there might be two MAPK cascades involving TaMAP4K in wheat, one associated with TaMAPK3 (87% similar to AtMAPK3), and the other associated with TaMAPK6 (90% similar to AtMAPK6). The protein–protein interaction network shown in Figure 7 involved two cascades: the first one was that TaMAPK6 might be activated by TaMAP2K11, which was in turn probably activated by TaMAP3K7; TaMAP3K7 could be activated by TaMAP4K10. On the other hand, TaMAP2K1/11 might be activated by TaMAP3K8, which was in turn probably activated by TaMAP4K10. These proteins phosphorylated TaMAP2K1/11, which in turn phosphorylated TaMAPK3 (Figure 7).

Notably, our results indicated that these *TaMAPK–TaMAP4K* cascade genes responded to various abiotic stresses. Expression analyses showed that *TaMAPK3*/*6* and *TaMAP4K10*/*24* were not only highly expressed during growth processes, but also responded to drought and heat stresses. Analyses of the synteny of orthologous segments of *TaMAPK3* and *6*, *TaMAP2K1*, *TaMAP3K7* and *8*, and *TaMAP4K10* displayed in Appendix A revealed that these six genes exhibit synteny across multiple phylogenetically related species. Furthermore, *TaMAPK3* is consistently organized in highly conserved genomic regions across all the species analyzed. These genes may be good candidates for genetic improvement of drought resistance and high grain yield in wheat.

## 3. Discussion

### 3.1. New Highlight: MAP4K in Wheat

In evolutionary studies, at least 10 protein kinases can be identified as MAP4K in the completely sequenced genomes of *Arabidopsis* and rice [26]. However, little is known about MAP4K in crops. Previous studies of the genome-wide analysis of the *Arabidopsis* MAP4K family members provided the basis for a comprehensive analysis of MAP4K in wheat. In this study, 25 putative TaMAP4K subfamily members exhibited strong similarity with the 10 MAP4Ks of *Arabidopsis*. The minimum and maximum amino acid sequence lengths of TaMAP4K are 146 and 1335, respectively (Appendix A). The TaMAP4Ks were named TaMAP4K1–25 (Appendix A). Based on the phylogenetic analysis of wheat and Arabidopsis MAP4K protein sequences, several highly orthologous proteins were identified between these two species, such as *TaMAP4K10*/*AtMAP4K3*, *TaMAP4K24*/*AtMAP4K4*, and *TaMAP4K9*/*AtMAP4K10* (Figure 4). In *Arabidopsis*, AtMAP4K3 (SIK1) was shown to regulate cell proliferation and expansion [26]. AtMAP4K4 associates with and regulates BIK1-mediated immunity in plants [39]. AtMAP4K10 (BLUE LIGHT SIGNALING1-BLUS1) encodes a putative Ser/Thr protein kinase that acts as a phototropin substrate and a major regulator of stomatal control to improve photosynthetic CO_2_ uptake in the presence of natural light [15]. Previously, it was reported that Pan and Smet screened 26 putative members of the TaMAP4K family in wheat. However, detailed analyses of the chromosomal localization, conserved domains, evolution, PPI network, and expression profiles of these genes were not conducted. Of the 26 putative members, *TraesCS5B02G196400* was assigned to *TaMAP3K25* [15]. Interestingly, the current study also found that three different subfamily genes (*TaMAP3K21*/*4K15*, *TaMAP3K22*/*4K10*, and *TaMAP3K23/4K21*) belonged to the same gene (Appendix A). These results indicate that there may be a certain degree of similarity between the TaMAP3K and TaMAP4K subfamilies.

Results from the current research showed that MAP3Ks could be activated through MAP4K phosphorylation in non-plant systems. For example, MAP4K2 of the germinal center kinases (GCKs) subfamily in mammals can activate the MEKK1 kinase cascade to activate Jun N-terminal kinase-JNK regulating cell signaling, immune cell activation, cell transformation, and cell migration [15]. In plants, MAP3Ks function downstream of RLKs and G-proteins [15,16,27], and no MAP4K cascade has been found to be involved so far. One possible explanation is that some MAP4K genes in plants lack functional kinase domains. A GxGxxG motif, for example, is important for nucleotide binding and is a vital element of the ATP-binding site with the third G replaced by A or S, which may lead to loss of activation function [16]. In this study, we noted that the Gs in 19 (76%) of the 25 TaMAP4K genes were substituted with As (Figure 1D and Appendix A). Moreover, HRD and DFG motifs, as well as an invariant lysine (K), played a critical role in catalysis [16]. However, our data showed that all 25 TaMAP4Ks contain three motifs: HRH, DFG, and invariant lysine (K) (Figure 1D and Appendix A). Of the MAP4K genes in wheat, TaMAP4K17 was the only one lacking the signature motif found in mammals (Figure 1D and Appendix A). Based on amino acid changes, these putative MAP4Ks may be kinase-inactive if they are mutant or lack all or part of the key kinase features.

### 3.2. TaMAPK–TaMAP4K Cascade Genes Are Conserved during Evolution

Syntenic analysis of the genomes facilitates the tracing of genomic inheritance across species and the identification of intra-species duplication events that are associated with pathway diversification [40]. The MAPK cascade is involved in multifaceted biological processes in eukaryotes, such as stress responses, growth and development, and immunity [2,36,41]. In this work, 234 TaMAPK genes were discovered in wheat and categorized into four subfamilies: MAPK (48 genes), MAP2K (17 genes), MAP3K (144 genes), and MAP4K (25 genes) (Appendix A). This distribution characteristic is similar to those reported in *Arabidopsis* (20 MAPK, 10 MAP2K, 80 MAP3K, and 10 MAP4K genes), *V. vinifera* (14, 5, 62, and 7, respectively), and *S. miltiorrhiza* (18, 9, 83, and 6 genes, respectively) [2,4,27]. Syntenic analyses indicate that the majority of *TaMAPK–TaMAP4K* genes have undergone purifying selection across the five ancestral species (Figure 5C, Appendix A), suggesting that these genes were conserved during the evolutionary process of these five ancestral lineages. This selection conservation may lead to the retention of common gene functions or characteristics across different ancestral species. In such scenarios, this conservation may endow the crucial roles of the *TaMAPK–TaMAP4K* genes in the survival or reproduction of the organisms during natural selection favoring the preservation of functional attributes.

The presence of introns can alter gene expression responses to environmental changes, resulting in a slowdown that impacts the duration between transcription and translation [2]. Variations in the gene structures in wheat are illustrated in Appendix A. The number of introns varied from 1 to 16 in *TaMAPK* genes, from 1 to 21 in TaMAP2Ks, from 1 to 23 in TaMAP3Ks, and from 3 to 22 in TaMAP4Ks (Appendix A). In *S. miltiorrhiza*, intron numbers varied from 1 to 11 in the *SmMAPK* genes, from 0 to 8 in the SmMAP2K genes, from 0 to 19 in the SmMAP3K genes, and from 0 to 23 in the SmMAP4K genes [4]. In *V. vinifera*, intron numbers varied from 3 to 15 in the *VvMPK* genes, from 2 to 8 in the VvMP2K genes, from 0 to 24 in the VvMP3K genes, and from 3 to 21 in the VvMP4K genes [5]. MAP2K genes do not have introns in some species such as *Brachypodium distachyon* and *Malus Domestica* [5]. Therefore, we found no significant correlation between intron number and previous reports of MAPK genes in other species in this study. In addition, all *TaMAPK–TaMAPK* cascade genes have introns. At the same time, we observed both segmental and tandem duplication events in *TaMAPK–TaMAP4K* cascade genes, with tandem duplication events being far rarer than segmental duplication events (Figure 2 and Appendix A). Tandem and segmental replication events can improve gene function diversity, allowing plants to adapt to a variety of environments. Hence, our findings suggest that *MAPK* cascade genes play an essential role in plant evolution.

To better understand the evolutionary relationships between the *TaMAPK–TaMAP4K* cascade proteins, evolutionary analysis was conducted based on the amino acid and genomic sequences of wheat (Figure 4). The TaMAPK, TaMAP2K, and TaMAP4K family can be divided into three groups. Among them, Group C has the most genes (Figure 4). TaMAP3K family genes are grouped into three subtypes: RAF, MEKK, and ZIK. These are the same as in *Arabidopsis* and other species, with the RAF group having the most genes (Figure 4 and Appendix A) [2]. Further, the conserved motifs of *TaMAPK–TaMAP4K* cascade genes are consistent with those previously reported in the green lineage [16]. The collinear relationship between wheat and *Arabidopsis* was similar to the phylogenetic results. Wheat and the five monocotyledonous plants have a significant level of collinearity, while collinearity with the dicotyledonous plant (*Arabidopsis*) was weakest, similar to the evolutionary relationship between monocotyledons and dicotyledons [42]. The new findings suggest that *TaMAPK–TaMAP4K* genes have been conserved throughout evolution.

### 3.3. The MAP4K10-MAP3K7/8-MAP2K1/2-MPK3/6 Cascade May Have an Important Function in the Drought/Heat Response as well as Growth and Development in Plants

The functions of *MAPK–MAP3K* cascade genes are well characterized in *plants* [35,43]. In this work, we analyzed the expression patterns of *TaMAPK–TaMAP4K* cascade genes in response to drought and heat stresses, as well as at different development stages through RNA-Seq data analysis (Figure 8 and Figure 9). These results suggested that *TaMAPK–TaMAP4K*s were involved in wheat abiotic stress as well as in growth and development signaling pathways. To discover the MAPK cascade pathway implicated in wheat abiotic stress tolerance, growth, and development, we performed qRT-PCR tests on 12 *TaMAPK–TaMAP4K* genes (Figure 10). Most of them contained *cis*-elements implicated in plant hormone signaling (MeJA, SA, IAA, ABA, GA), defense, and stress responses (MBS, LTR, ARE) (Appendix A). *MAPK* genes are important in responses to abiotic stresses as well as in the control of growth and development in plants. Previous studies showed that overexpression of *MdRaf5* (MAP3K) in *M. domestica* enhanced drought resistance in *Arabidopsis* [44]. Drought induced the expression of the *FvMAPK5* and *FvMAPK8* genes, which were orthologs of the *AtMAPK3* gene [25]. Moreover, transcription levels of *FvMAPK3* and *FvMP2K1*/*3*/*6*/*7* were significantly upregulated under heat stress [25]. *FtMAPK4/5* genes as well as the Raf subfamily, *FtMAP3K4*/*24*/*32*/*53*, were highly upregulated under drought stress in *F. tataricum*, whereas *FtMAP5*/*8* and *FtMAP2K1* were highly upregulated following treatment at 40 °C [42]. In mulberry, *MnMAPK*1/5/6/9 were upregulated while *MnMAPK*2/3/8/10 were downregulated under 40 °C stress [43]. MAPKs can function in the cytoplasm as well as translocate to the nucleus to activate other proteins or transcription factors, and the MAPK cascade may regulate nitrogen assimilation [45]. In the *Green Alga Chlamydomonas reinhardtii*, the genetic expression of MAPKs and MAPKKs is subtly regulated by nitrogen, with the MAP3K genes RAF14 and RAF79 showing strong ammonium inhibition, indicating their potential key roles in the regulation of nitrogen assimilation [45]. We also found that some *MAP4K* genes have been identified in plant genomes through kinase domain and evolutionary analysis. For example, the ectopic expression of *ZmSIMK1* in *Zea mays* enhanced drought resistance in tobacco [44]. *ScMAP4K1* was mainly expressed during the development of ovules, seeds, and fruits in *Solanum chacoense* [28]. *BnMAP4Ka1* and *BnMAP4Ka2* played important roles at different stages of embryogenesis, in roots, and in flower buds and siliques [22]. In *Arabidopsis*, *AtMAPK3* and *AtMAPK36* participated in the response to abiotic stress [4]. Results from this study showed that *TaMAPK3/6/14, TaMAP2K11*, *TaRaf76*, and *TaMAP4K3/10* were significantly upregulated after drought and heat treatment, with *TaMAP3K7* responding specifically to drought and combined drought and heat (6 h) stress, and *TaMAPK3*/*6*, *TaMAP2K1*/*11*, *TaMAP3K1/8*, and *TaMAP4K10*/*24* being highly expressed in the roots, stems, leaves, spikes, and grains (Figure 6 and Figure 8). Notability, *TaMAPK3*/*6* were homologous to *AtMAPK3/6*, *FvMAPK5*/*8*, *FtMAPK4*/*5*, and *MnMAPK*1/5/6. These genes belong to the same subfamily and all of them contain not only the same TEY domain but also an ABA *cis*-element (Figure 1, Figure 6 and Appendix A, and Appendix A). *TaRaf76* was homologous to *FtMAP3K4*/*32*/*53*. Five *MAPK* cascade genes belong to the Raf subfamily and have MBS and MeJA *cis*-element (Figure 1 and Appendix A). *TaMAP4K3*/*10*/*24* were highly homologous to *ZmSIMK1*, *BnMAP4Ka1*, and *BnMAP4Ka2*, and these genes contained the same domains (Glycine-rich, HRD, DFG motif, and signature motif) [28,46], and *TaMAP4K24* were highly expressed in the roots (RTLS and RMS) and grains (G2DAAs and G14DAAs) (Figure 1D, Figure 9, Figure 10 and Appendix A, and Appendix A). *TaMAP4K3* and *10* were highly upregulated by drought (6 h), heat, as well as combined drought and heat stresses, indicating they have the same function as *ZmSIMK1* (Figure 10 and Appendix A). These findings imply that the activities of these homologous genes are likely conserved in different plants, with distinct roles in plant growth and a diversified environment.

A typical MAPK cascade module consists of three kinase components. The CRLK1/2-MEKK-MAP2K1/2-MAPK4 pathway, for example, positively regulates cold stress responses [47]. The MAP3K17/18-MKK3-MAPK1/2/7/14 pathway regulated stress responses in *Arabidopsis* [9]. The GhMAP3K62–GhMKK16–GhMAPK32 pathway participated in drought stress responses in cotton [13,48]. Typically, MAP3Ks can be activated through MAP4K phosphorylation in non-plant systems. However, the MAP3K in plants is generally RLKs or G proteins. In this study, the bioinformatics analysis and qRT-PCR results suggest that TaMAP4K10, TaMAP3K7, TaMAP2K1/11, TaMAPK3/6, TaMAP4K10, TaMAP3K8, TaMAP2K1/11, and TaMAPK3 undergo protein–protein interactions (Figure 7 and Appendix A). Moreover, PPI network analysis indicates that TaMAP2K11 has strong interactions with other proteins such as TaMAP3K1/7/8 and TaMAPK3/6 (Figure 7 and Appendix A). Under drought stress, the expression levels of *TaMAP4K10*, *TaMAP3K7*, *TaMAP2K1*/*11*, and *TaMAPK3*/*6* were significantly enhanced; these genes contain *cis*-elements associated with drought, ABA, and MeJA stress responses (Figure 6 and Figure 10, and Appendix A). *TaMAP3K7*, *TaMAP2K11*, and *TaMAPK6* genes contain a *cis*-element that binds to MYB and contributes to the drought stress response (Figure 6 and Appendix A). As a result, we anticipate that drought stress may activate the TaMAP4K10–TaMAP3K7–TaMAP2K11–TaMAP2K6 pathway. This pathway boosts drought resistance in wheat by interacting with MYB transcription factors. *TaMAP4K10*, *TaMAP3K8*, *TaMAP2K1/11*, and *TaMAPK3* were highly expressed in roots and stems, and they contain *cis*-elements associated with auxin and ABA responses (Figure 6 and Figure 10, and Appendix A). During plant growth, the TaMAP4K10–TaMAP3K8–TaMAP2K1/11-TaMAP2K3 pathway may be activated to promote plant growth. The interspecific differences in the structure of the MAPK cascade and the interspecific similarity in *MAPK* gene function explain the variability and conservation of gene evolution. Syntenic analysis also revealed that *MAP4K10*, *MAP3K7* and *MAP3K8*, *MAP2K1*, *MAPK3*, and *MAPK6* have undergone purifying selection, indicating selective conservation of these six genes across different species during the course of evolution (Figure 5C, Appendix A). This conservation has resulted in the retention of common gene functions or characteristics that are of significant importance for the growth, development, and stress responses in these six species.

## 4. Materials and Methods

### 4.1. Plant Materials and Treatments

Bread wheat (*Triticum aestivum* L. cv. Fielder) was acquired from the CSIRO Plant Industry laboratory of Prof. Xue Gangping [37]. For sterilization, wheat seeds were immersed in a 15–20% sodium hypochlorite (NaClO) solution for 25–30 min, then rinsed in deionized water. The seeds were germinated on moist paper towels, cultured at 4 °C for 5 days, and then transferred to 12 °C for another 5 days to prepare an adequate amount of wheat seedlings. The seedlings were divided into four groups (each with at least 50 uniformly growing seedlings). The first group included wheat seedlings sampled from different tissues and at different time points (seedling roots: root_10 days (SR), roots at three leaves stage: root_13 days (RTLS), roots at meiosis stage: root_39 days (RMS), stem at the 1 cm spike stage: stem_30 days (S1S), stems at the two nodes stems stage: stem_32 days (STNS), stems at anthesis stage: stem_65 days (SATS), seedling leaves: leaf_10 days (SL), leaves at three tillers stages: leaf_23 days (LTTS), leaves at 2 days after anthesis stage: leaf_71 days (L2DAAs), spikes at two nodes stem stage: spike_32 days (SPTNS), spikes at the meiosis stage: spike_39 days (SPMS), spikes at anthesis stage: spike_65 days (SPAS), grains at 2 days after anthesis stage: grain_71 days (G2DAAs), grains at 14 days after anthesis stage: grain_75 days (G14DAAs), grains at 30 days after anthesis stage: grain_85 days (G30DAAs)). The second group included root systems of seedlings subjected to 0, 1, and 6 h of drought stress via immersion in a half-strength Hoagland solution containing 25% PEG6000. The third group included wheat seedlings immersed in Hoagland solution and subjected to 0, 1, and 6 h of heat stress at 37 °C in a growth chamber. The fourth group included root systems of seedlings subjected to simultaneous drought and heat stress via immersion in a half-strength Hoagland solution containing 25% PEG6000 for 0, 1, and 6 h. The 0 h-treated samples served as a control. The growth conditions for the first and second groups of seedlings were: temperature at 22 °C/18 °C, 16 h of light, and 8 h of darkness. The growth conditions for the third and fourth groups of seedlings were: 16 h of light and 8 h of darkness. All experimental samples were collected and sampled by the author personally, with three biological replicates sampled at different stress treatment time points or different stages of tissue development within each group [32,49]. The samples were frozen in liquid nitrogen and kept at −80 °C until use.

### 4.2. Genome-wide Identification of TaMAPK–TaMAP4K Family Members in Wheat

To comprehensively identify *TaMAPK*–*TaMAP4K* members, the reference genome and corresponding annotated datasets and protein sequence information for MAPK–MAP4K family members in *Triticum aestivum* (wheat, AABBDD), *Triticum urartu* (*T. urartu*, AA), *Aegilops speltoides* (*A. speltoides*, BB), *Triticum dicoccoides* (*T. dicoccoides*, AABB), *Aegilops tauschii* (*A. tauschii*, DD), *Triticum turgidum* (*T. turgidum*, AABB), *Oryza sativa Japonica Group* (rice), and *Arabidopsis* were obtained from the Ensembl Plants database (https://plants.ensembl.org/index.html, 23 September 2023) and ScienceDB (https://www.scidb.cn/en/s/VFrU7z, 1 October 2023) [39,50,51,52]. The data were used to create a local protein database. MAP4K protein sequences of *Arabidopsis* and the known MAPK, MAP2K, and MAP3K amino acid sequences collected from wheat (54 MAPK, 18 MAP2K, 155 MAP3K) were used as queries to search against the MAPK–MAP4K amino acid sequences of wheat mentioned above using the BLASTP program [1]. The available *Arabidopsis MAPK*–*MAP4K* genes were used to develop a Hidden Markov Model (HMM) profile using the hmm-build tool incorporated in HMMER 3.0 (http://hmmer.org/,4 October 2023), and the HMM profile was then used to search for wheat MAPK–MAP4K proteins using the hmmsearch tool embedded in HMMER. The HMMER hits were then combined with the BLASTP findings and manually edited to eliminate any redundant sequences. Then, the number of amino acids, grand average of hydropathicity (GRAVY), isoelectric points (pIs), and molecular weights (MWs) of the *TaMAPK–TaMAP4K* genes were predicted using ProtParam (http://web.expasy.org/protparam/,19 October 2023). Subcellular localization prediction of *TaMAPK–TaMAP4K* proteins was carried out using the Cell-PLoc 2.0 server (http://www.csbio.sjtu.edu.cn/bioinf/Cell-PLoc-2/, 25 October 2023) [53]. Transmembrane domains were predicted using TMHMM (https://dtu.biolib.com/DeepTMHMM, 30 October 2023).

### 4.3. Chromosome Locations, Gene Structure, Multiple Alignment, Conserved Motifs, and Phylogenetic Tree Analyses of TaMAPK–TaMAP4K Cascade Proteins

The annotation file of the wheat genome was searched using TBtools to visualize chromosome locations and gene structures of the *TaMAPK–TaMAP4K* cascade members. The predicted *TaMAPK*–*TaMAP4K* genes were submitted to the MEME (https://meme-suite.org/meme,6 September 2023) for analysis to identify their conserved motifs. Protein sequences containing the MAPK domain were used to identify target genes. The *TaMAPK–TaMAP4K* protein sequences were compared using default parameters in ClusterW [1]. A phylogenetic tree was built in the MEGA v11.0.10 software using the Neighbor-Joining method, with 1000 bootstrap replicates and default parameters [1].

### 4.4. Cis-Element, Duplication Events, and Collinearity of TaMAPK–TaMAP4K

The ATG upstream sequences (1500 bp) of *TaMAPK*–*TaMAP4K* genes were uploaded to the PlantCARE database (http://bioinformatics.psb.ugent.be/webtools/plantcare/html/, 24 September 2023) to analyze various *cis*-acting elements [17,24,36,42,53]. The evolutionary relationships between hexaploid wheat and the five representative species were investigated using TimeTree (http://www.timetree.org/, 30 September 2023) [54]. Furthermore, a comprehensive syntenic examination of *MAPK*, *MAP2K*, *MAP3K*, and *MAP4K* genes in wheat and five representative species, namely *T. urartu, A. speltoides, T. dicoccoides*, *A. tauschii*, and *T. turgidum*, was carried out using the TBtools. This was followed by computing the synonymous (Ks) and non-synonymous (Ka) substitutions in the duplicated gene pairs in order to further assess duplication events. The time (T) of duplication in millions of years (Mya) was estimated using the formula T = Ks/2λ × 10^−6^ Mya, (λ = 6.5 × 10^−9^) [55].

### 4.5. Protein–Protein Interaction (PPI) Network Analysis

Interaction networks of *TaMAPK–TaMAP4K* were analyzed using STRING (https://string-db.org/cgi/input.pl, 11 September 2023). The wheat database was used as a reference (https://plants.ensembl.org/index.html, 22 September 2023). The predictions were stored in TSV format and then loaded into the Cytoscape V3.10.1 program [56] for display.

### 4.6. Expression Profiles and qRT-PCR Analysis of TaMAPK–TaMAP4K Genes

A total of 234 RNA sequences from the Wheat Expression Browser (http://www.wheat-expression.com/, 11 October 2023) were used to investigate the differential expression of *TaMAPK*–*TaMAP4K* genes in various organs (roots, stems, leaves, spikes, and grains) [50]. Wheat plants’ responses to drought and heat stresses were analyzed (Appendix A). A total of 234 *TaMAPK*–*TaMAP4K* genes were examined at 15 different growth stages: SR, RTLS, RMS, S1S, STNS, SATS, SL, LTTS, L2DAAs, SPTNS, SPMS, SPAS, G2DAAs, G14DAAs, and G30DAAs [32,49,53]. A total of 180 *TaMAPK*–*TaMAP4K* genes were tested under 6 different treatment conditions: 1 h drought-(1 h-d), 6h drought-(6 h-d), 1h heat-(1 h-h), 6 h heat-(6 h-h), 1h drought and heat-(1 h-dh), and 6 h drought and heat-(6 h-dh) [32,49,53]. Total RNA was isolated from tissues of plants at various developmental stages and from stressed-treated plants using Trizol reagent (Vazyme, Nanjing, China) based on the manufacturer’s instructions. First-strand cDNAs were generated using the SuperMix (Vazyme, Nanjing, China) and kept at −20 °C. The RNA samples were treated with deoxyribonuclease (DNAse). The quantitative real-time polymerase chain reaction (qRT-PCR) primers used in this study were created with primer 3 plus (https://www.primer3plus.com/, 16 October 2023). We designed specific primers for each gene’s cDNA sequence (Appendix A). TaRP15 was an internal reference gene [53]. Relative gene expression levels were calculated using the 2^−ΔΔCT^ method, and statistical analyses were conducted using *t*-tests to assess the significance of differences in the data [53]. All experiments were performed using three replicates. All the samples were frozen in liquid nitrogen immediately and kept at −80 °C until use.

## 5. Conclusions

In the study reported here, we identified 234 members of the MAPK–MAP4K gene family in the wheat genome. They included 48 MAPKs, 17 MAP2Ks, 144 MAP3Ks, and 25 MAP4Ks. The MAPK–MAP4K family genes were first validated in wheat. We conducted a systematic analysis of the evolution, domain conservation, interaction networks, and expression profiles of the *TaMAPK–TaMAP4K* kinase family members. The 234 *TaMAPK–TaMAP4K*s were distributed on each of the 21 wheat chromosomes and Un. The 25 TaMAP4K family members possessed the conserved glycine-rich motif, invariant lysine (K) motif, HRD motif, DFG motif, and signature motif of MAP4K. Furthermore, TaMAP4K contains VGTPxWMAPEV sequences in its kinase domain that are highly similar to yeast Ste20-like MAP4Ks in the GCK subfamily. Both the N- and C-terminal parts of these genes are highly similar to those of AtMAP4Ks and BnMAP4Ks. According to public RNA-seq data, *TaMAPK3*/*6*/*14*/*36*, *TaMAP2K1*/*2*, *TaMAP3K7*, *TaRaf14/68/113, TaZIK5*/*8*, and *TaMAP4K3*/*4* were upregulated under drought, heat, or combined drought and heat condition. On the contrary, the expressons of *TaMAPK17*/*46*, *TaMAP2K11*/*15*, *TaMAP3K21/23*, *TaRaf14/68/113, TaZIK2/9/10*, and *TaMAP4K16*/*19* were suppressed. *TaMAPK3*, *6*, and *TaMAP4K10*/*24* were strongly expressed not only throughout the growth and development stages but also in response to drought or heat stress. The bioinformatics analyses and qRT-PCR results suggested that wheat may activate the MAP4K10–MEKK7–MAP2K11–MAPK6 pathway to increase drought resistance in wheat, and the MAP4K10–MAP3K8–MAP2K1/11-MAPK3 pathway may be involved in plant growth. In general, our work identified members of the MAPK–MAP4K cascade in wheat and profiled their potential roles during responses to abiotic stresses and plant growth based on their expression patterns.

## Figures and Tables

**Figure 1 plants-13-00941-f001:**
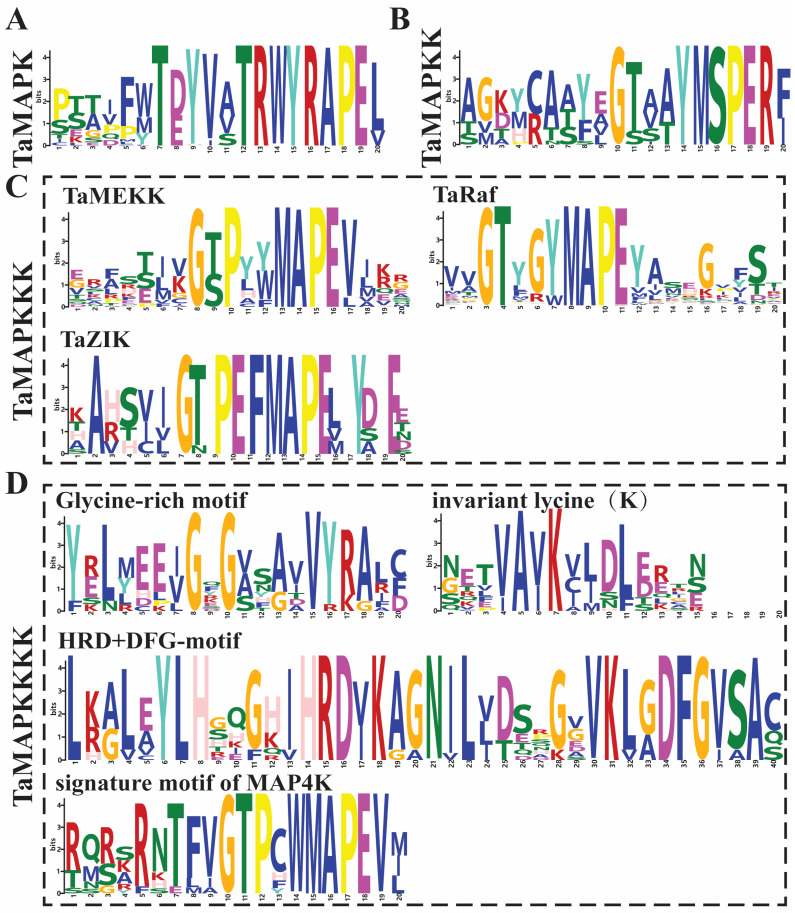
Conserved motifs of *TaMAPK–TaMAP4K* family genes in wheat. (**A**): the conserved motifs in MAPK proteins. (**B**): the conserved motif in MAP2K proteins. (**C**): the conserved motif in MAP3K (MEKK, Raf, and ZIK) proteins. (**D**): the conserved motif in MAP4K (Glycine-rich, invariant lysine (K), HRD, DFG, and signature motif of MAP4K) proteins. The height of symbols within the stack indicates the relative frequency(relative proportion and conserved property) of each amino at that position.

**Figure 2 plants-13-00941-f002:**
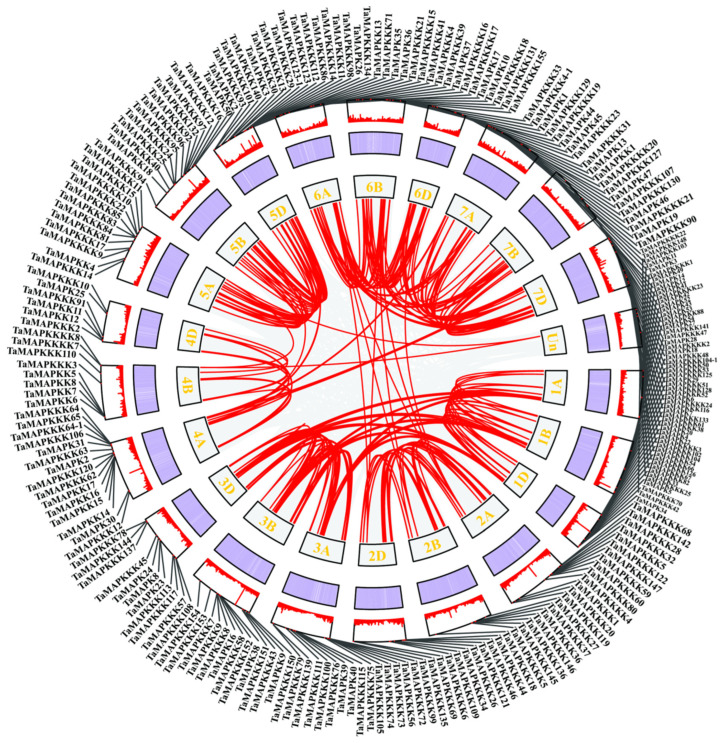
Schematic representations of *TaMAPK*–*TaMAP4K* cascade gene duplications. The gray line in the background indicates the homolinear pair in the whole wheat genome. Red lines indicate the collinear *TaMAPK*–*TaMAP4K* gene pairs. The chromosome number is displayed on each chromosome. The heatmap and line chart in the outer square represent gene density.

**Figure 3 plants-13-00941-f003:**
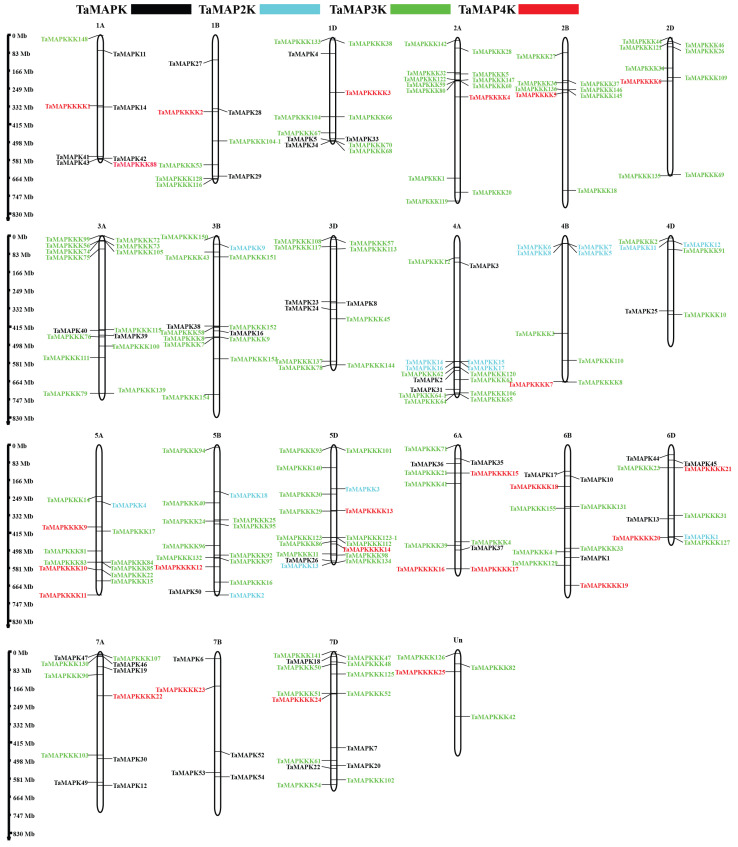
Chromosomal locations of *TaMAPK–TaMAP4K* cascade genes. The chromosome number is indicated at the top and the scale on the left represents the chromosome size. Black indicates TaMAPK, blue indicates TaMAP2K, green indicates TaMAP3K, and red indicates TaMAP4K.

**Figure 4 plants-13-00941-f004:**
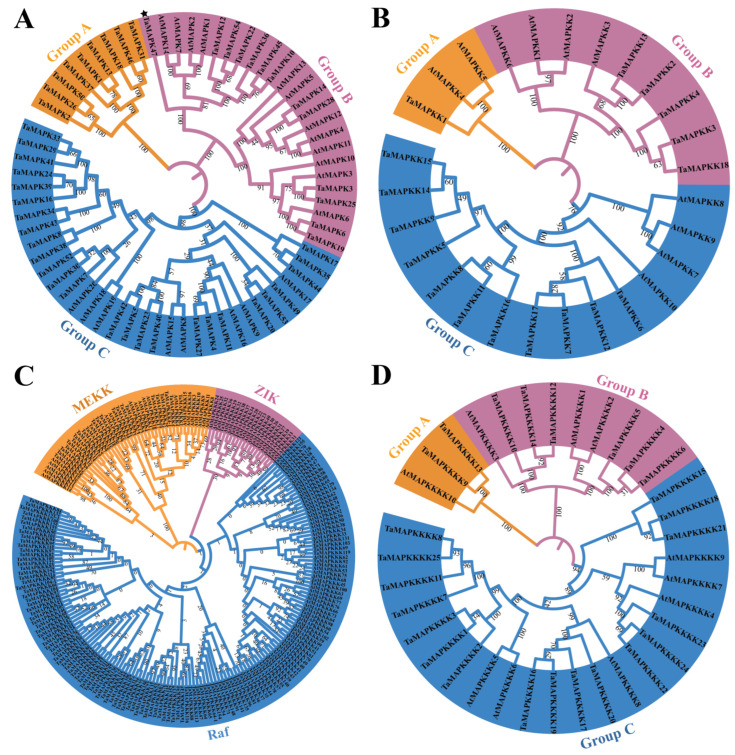
Phylogenetic analyses of *TaMAPK–TaMAP4K* cascade genes in wheat and *Arabidopsis.* (**A**): phylogenetic tree of the MAPK gene family, TaMAPK47 in Group B was denoted by star. (**B**): phylogenetic tree of the MAP2K gene family. (**C**): phylogenetic tree of the MAP3K gene family (TaMEKK, TaZIK, and TaRaf). (**D**): phylogenetic tree of the MAP4K gene family.

**Figure 5 plants-13-00941-f005:**
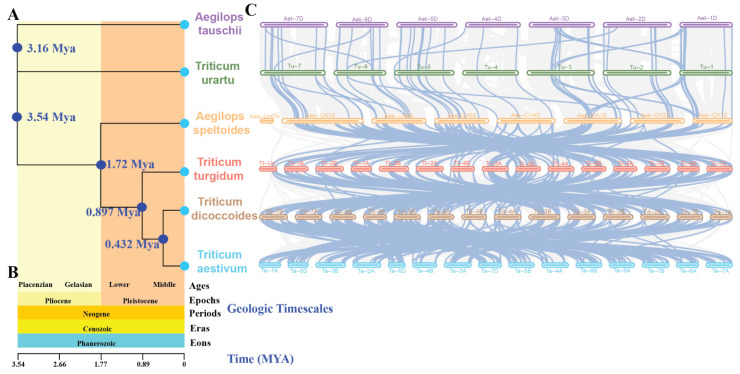
(**A**): A phylogenetic tree based on whole-genome analysis of 6 species was constructed for syntenic analysis of *A. tauschii*, *T. urartu*, *A. speltoides*, *T. turgidum*, *T. dicoccoides*, and *T. aestivum*. The estimated divergence times for each node are depicted in blue. (**B**): The macro-geological timescale of the evolutionary events of 6 species, including *A. tauschii*, *T. urartu*, *A. speltoides*, *T. turgidum*, *T. dicoccoides*, *and T. aestivum* encompassing Ages (Piacenzian, Gelasian, Lower, Middle), Epochs (Pliocene, Pleistocene), Periods (Neogene), Eras (Cenozoic), and Eons (Phanerozoic). (**C**): Syntenic analysis was conducted to compare the *MAPK–MAP4K* cascade genes in wheat with those in the five ancestral species.

**Figure 6 plants-13-00941-f006:**
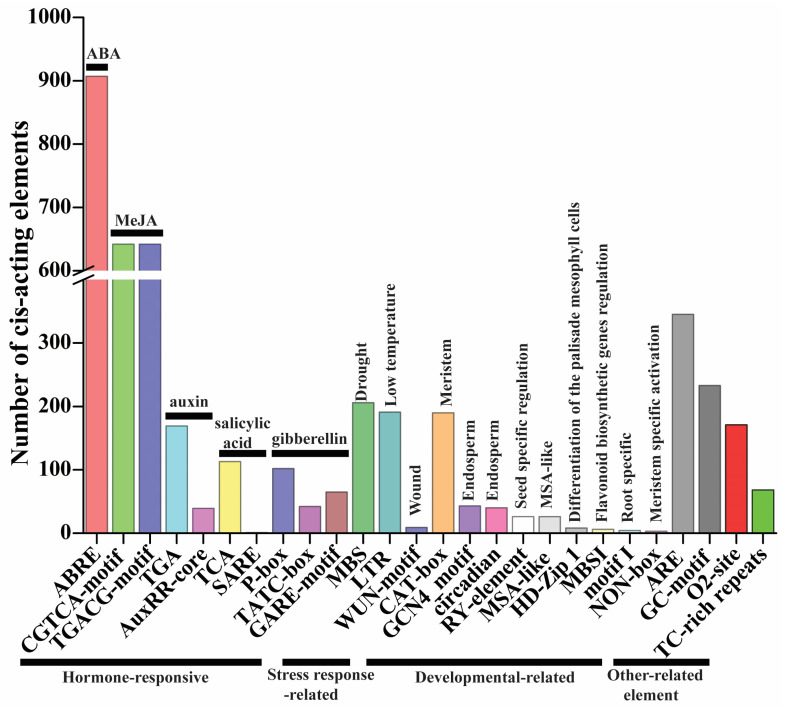
The number of *cis*-regulatory elements in the promoter regions of *TaMAPK–TaMAP4K* cascade genes. The names of the *cis*-regulatory elements are listed at the bottom of the image, and the corresponding functional annotations are listed above the image.

**Figure 7 plants-13-00941-f007:**
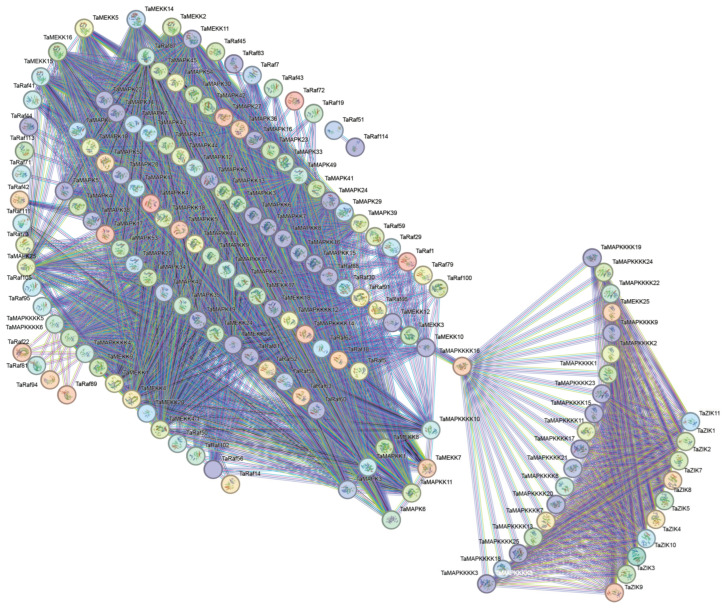
Protein–protein interaction network diagram of *TaMAPK–TaMAP4K* in wheat. The network nodes represent proteins. The edges represent protein–protein associations. Colored nodes: query proteins and first shell of interactors. White nodes: second shell of interactors. Empty nodes: proteins with unknown 3D structures. Filled nodes: 3D structures are known or predicted. The more nodes one node is connected to, the higher its degree value.

**Figure 8 plants-13-00941-f008:**
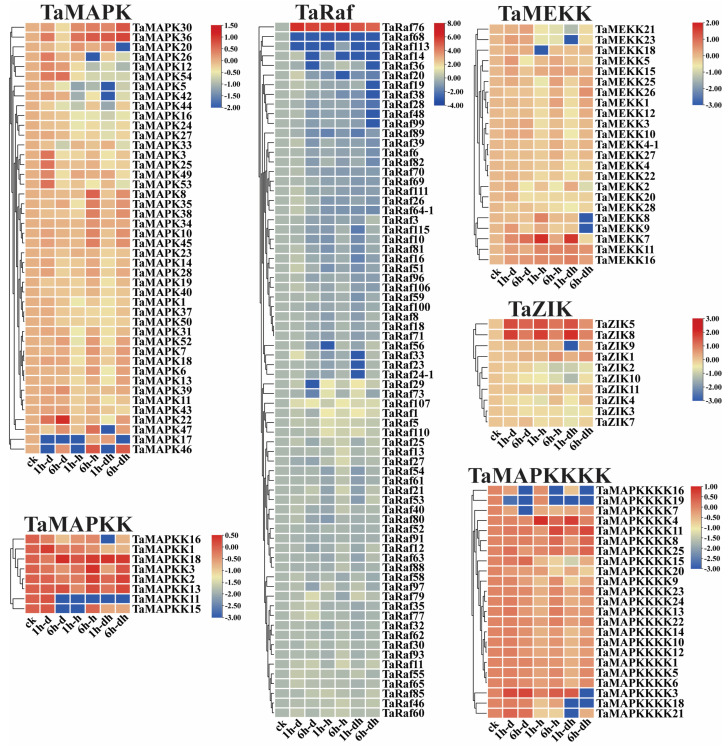
Clustering of the expression patterns of the 234 *TaMAPK*–*TaMAP4K* cascade genes in wheat under drought, heat, and combined drought and heat stresses (1 h drought–(1 h–d), 6 h drought–(6 h–d), 1h heat–(1 h–h), 6 h heat–(6 h–h), 1h drought and heat–(1 h–dh), 6 h drought and heat–(6 h–dh)). Log10 logarithmic transformation treatment was applied to gene expression analysis. High levels of gene expression are in red and low levels of gene expression are shown in blue.

**Figure 9 plants-13-00941-f009:**
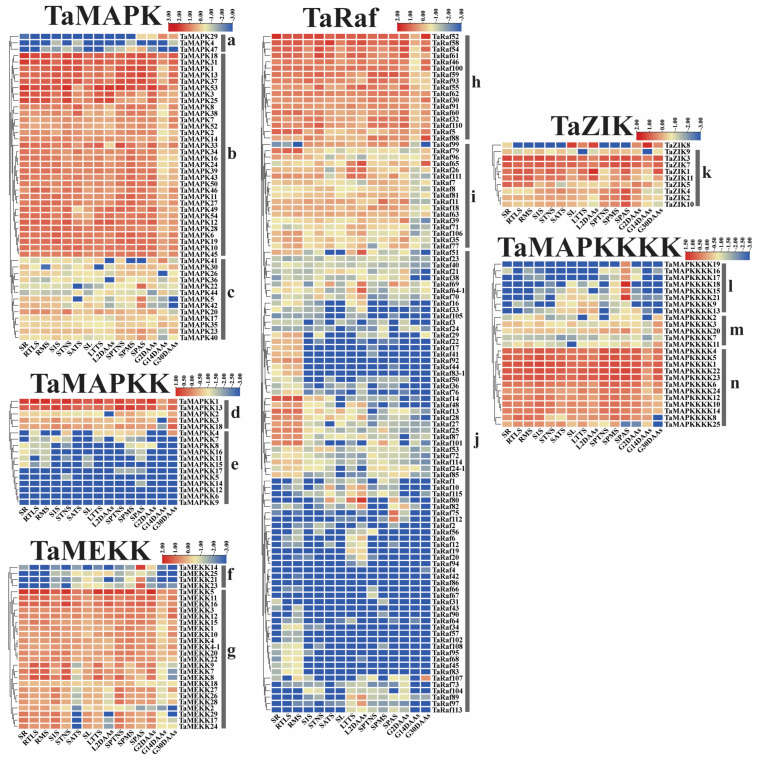
Clustering of the expression patterns of the 234 *TaMAPK–TaMAP4K* cascade genes in wheat at 15 different developmental stages (seedling roots: root_10 days (SR), roots at three-leaf stage: root_13 days (RTLS), roots at meiosis stage: root_39 days (RMS), stems at the 1 cm spike stage: stem_30 days (S1S), stems at the two nodes stems stage: stem_32 days (STNS), stems at anthesis stage: stem_65 days (SATS), seedling leaves: leaf_10 days (SL), leaves at three tillers stage: leaf_23 days (LTTS), leaves at 2 days after anthesis stage: leaf_71 days (L2DAAs), spikes at two nodes stems stage: spike_32 days (SPTNS), spikes at the meiosis stages: spike_39 days (SPMS), spikes at anthesis stage: spike_65 days (SPAS), grains at 2 days after anthesis stage: grain_71 days (G2DAAs), grains at 14 days after anthesis stage: grain_75 days (G14DAAs), grains at 30 days after anthesis stage: grain_85 days (G30DAAs)). Log10 logarithmic transformation treatment was applied when analyzing gene expression. Red indicates high gene expression and blue indicates low expression. The letters a–n indicate the groups of genes with different expression levels.

**Figure 10 plants-13-00941-f010:**
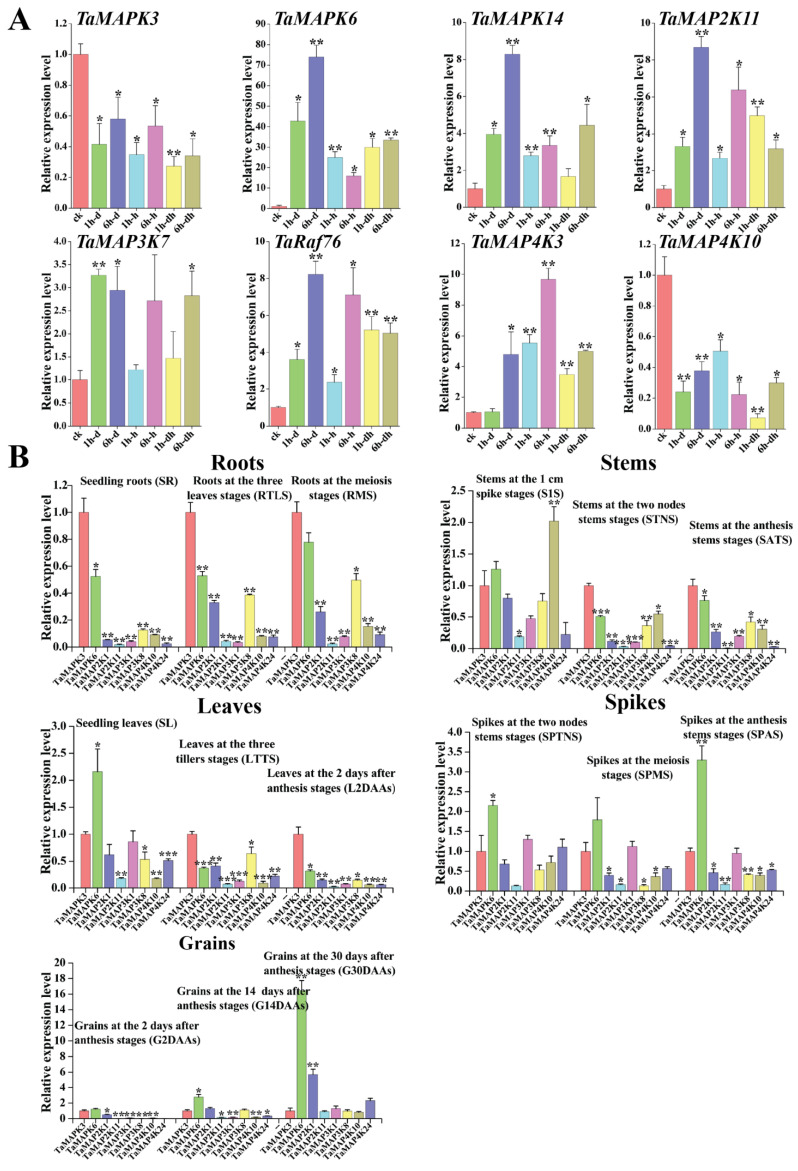
Quantitative RT-PCR analysis of 16 *TaMAPK–TaMAP4K* genes. (**A**): Relative expression levels of 8 *TaMAPK–TaMAP4K* genes under 3 different stress treatments ((Check-CK), 1 h drought-(1 h-d), 6 h drought-(6 h-d), 1h heat-(1 h-h), 6 h heat-(6 h-h), 1h drought and heat-(1 h-dh), 6 h drought and heat-(6 h-dh)). (**B**): Relative expression levels of 8 *TaMAPK–TaMAP4K* genes at 15 developmental stages (seedling roots: root_10 days (SR), roots at three-leaf stage: root_13 days (RTLS), roots at meiosis stage: root_39 days (RMS), stems at the 1 cm spike stage: stem_30 days (S1S), stems at the two nodes stem stage: stem_32 days (STNS), stems at anthesis stage: stem_65 days (SATS), seedling leaves: leaf_10 days (SL), leaves at three tillers stage: leaf_23 days (LTTS), leaves at 2 days after anthesis stage: leaf_71 days (L2DAAs), spikes at two nodes stems stage: spike_32 days (SPTNS), spikes at the meiosis stage: spike_39 days (SPMS), spikes at anthesis stage: spike_65 days (SPAS), grains at 2 days after anthesis stage: grain_71 days (G2DAAs), grains at 14 days after anthesis stage: grain_75 days (G14DAAs), grains at 30 days after anthesis stage: grain_85 days (G30DAAs)). *TaMAPK3* gene expression data from different stages and tissues were used as the control. Vertical bars indicate standard deviations (* *p* < 0.05, ** *p* < 0.01, *** *p* < 0.001, Student’s *t*-text).

## Data Availability

The original data presented in the study are publicly available. All information on MAPK–MAP4K sequences of wheat, *Triticum aestivum* (wheat, AABBDD), *Triticum urartu* (*T. urartu*, AA)*, Aegilops speltoides* (*A. speltoides*, BB)*, Triticum dicoccoides* (*T. dicoccoides*, AABB), *Aegilops tauschii* (*A. tauschii*, DD), *Triticum turgidum* (*T. turgidum*, AABB), *Oryza sativa Japonica Group* (rice), and *Arabidopsis* were acquired from *Ensembl Plants* database (http://plants.ensembl.org/Triticum_aestivum/Info/Index) and ScienceDB (https://www.scidb.cn/en/s/VFrU7z). Subcellular localization data for *TaMAPK–TaMAP4K* were acquired from the Cell-PLoc 2.0 server (http://www.csbio.sjtu.edu.cn/bioinf/Cell-PLoc-2/). Transmembrane domain data were acquired from TMHMM (https://dtu.biolib.com/DeepTMHMM). The RNA-Seq data used in this study were acquired from the Wheat Expression Browser (http:// www.wheat-expression.com/) under accession number SRP045409.

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
