# Peer review of "Bioinformatic Identification and Expression Analyses of the MAPK–MAP4K Gene Family Reveal a Putative Functional MAP4K10-MAP3K7/8-MAP2K1/11-MAPK3/6 Cascade in Wheat (Triticum aestivum L.)"

_plants, 2024, doi:10.3390/plants13070941_

Round 1
Reviewer 1 Report (New Reviewer)
Comments and Suggestions for Authors
Dear Authors,
In this paper, the authors investigate the MAPK protein family in wheat and its relationship water stress and growth. Special emphasis is placed on the MAP4K family, which has been poorly characterized. I consider the article to be novel in this regard. The paper highlights the vast amount of information presented, mostly bioinformatic, with experimental sections featuring qRT-PCR experiments. The information is thoroughly discussed, although I would suggest rearranging the order of the results. And very importantly, improve the materials and methods sections.
Majors:
In regard to the qRT-PCR, In the materials and methods section, it's not clear at all how the sampling was done, did the authors collect the samples themselves or obtain the data from other papers results? This aspect should be clearly explained in this section. How many plants were used? What were the exact cultivation conditions were they in a greenhouse?
I consider that the introduction is quite rigorous and well-written; I just propose a couple more suggestions.
L51: Besides this, it would also be convenient to indicate that, in Arabidopsis, MAPK6 has also been implicated in the regulation of the production of the second messenger nitric oxide through the regulation of nitrate reductase doi.org/10.1105/tpc.109.072959
In the results, I believe it would be appropriate for the authors to provide a structural explanation (conserve motif) of the basis for determining which MAPK belongs to one family or another, that is, to indicate specifically the molecular markers that distinguish them. The MAPK domain is TxY (T loop)……
Figure 1: I advise the authors to color-code each of the families differently. This would allow the chromosomal distribution of the different families to be seen at a glance more easily.
I believe this section is where the results should have started being presented, rather than the chromosomal localization. This part is scientifically more relevant, in my opinion. I strongly recommend the authors to change the order of the results, presenting this data first. Meaning, I recommend that section 2.3 becomes section 2.1.
L247-L249: This reasoning is more appropriate in the discusion.
L265-L267 This reasoning is more appropriate in the discusion.
Figure 4. In the materials and methods, I don't see how the phylogenetic study was conducted, this needs to be corrected, and it's very important to indicate in Fig. 4 the bootstrap values and the number of replicates.
L334: “The 1500bp promoter sequence” Provide an explanation of why a size of 1500 bp was chosen and not another. Please
Fig 10B: Specify the meaning of the abbreviations for the 15 developmental stages.
L504: It has been discovered that MAPKs are also involved in nitrogen metabolism doi:10.3390/ijms21103417
Author Response
Dear Authors,
In this paper, the authors investigate the MAPK protein family in wheat and its relationship water stress and growth. Special emphasis is placed on the MAP4K family, which has been poorly characterized. I consider the article to be novel in this regard. The paper highlights the vast amount of information presented, mostly bioinformatic, with experimental sections featuring qRT-PCR experiments. The information is thoroughly discussed, although I would suggest rearranging the order of the results. And very importantly, improve the materials and methods sections.
Majors:
In regard to the qRT-PCR, In the materials and methods section, it's not clear at all how the sampling was done, did the authors collect the samples themselves or obtain the data from other papers results? This aspect should be clearly explained in this section. How many plants were used? What were the exact cultivation conditions were they in a greenhouse?
Response: Thank you very much for your dedicated review and constructive suggestions. All sections in the manuscript requiring supplementation and revision have been indicated using blue highlighting.
For the qRT-PCR analyses, the plant samples and data were collected by us, instead of sourcing from other papers. We further supplemented the related materials and methods, including the seed treatment methods, sampling procedures, the number of plants analyzed, and the growth conditions of plants. Line 498-518
The revision are highlighted in blue.
I consider that the introduction is quite rigorous and well-written; I just propose a couple more suggestions.
L51: Besides this, it would also be convenient to indicate that, in Arabidopsis, MAPK6 has also been implicated in the regulation of the production of the second messenger nitric oxide through the regulation of nitrate reductase doi.org/10.1105/tpc.109.072959
Response: We have supplemented. Line 48-49
In the results, I believe it would be appropriate for the authors to provide a structural explanation (conserve motif) of the basis for determining which MAPK belongs to one family or another, that is, to indicate specifically the molecular markers that distinguish them. The MAPK domain is TxY (T loop)……
Response: We have indicated specifically the molecular markers that distinguish them in paper, as well as Figure 3 and Figure S2-7. The revision are highlighted in blue.
The MAPK domain is T(E/D)YVxTRWYRAPE(L/V) (Line 110-111).
The MAP2K domain are VGTxxYMSPER and the putative S/T-x5-S/T motif (Line 111-112).
The MAP3K domain are MEKK (G(T/S)Px(W/Y/F)MAPEV), RAF (GTxx(W/Y)MAPE), and ZIK (GTPEFMAPE(L/V)Y) (Line 114-115).
The MAP4K domain are glycine-rich motif (GxGxxG/A), invariant lycine (K) motif, HRD motif, DFG motif, and signature motif of MAP4K (VGTPxWMAPEV) (Line 119-120).
Figure 1: I advise the authors to color-code each of the families differently. This would allow the chromosomal distribution of the different families to be seen at a glance more easily.
Response: We have applied distinct color codes to the four families, TaMAPK, TaMAP2K, TaMAP3K, and TaMAP4K, whereby black indicates TaMAPK, blue indicates TaMAP2K, green indicates TaMAP3K, and red indicates TaMAP4K. Line 179-181
I believe this section is where the results should have started being presented, rather than the chromosomal localization. This part is scientifically more relevant, in my opinion. I strongly recommend the authors to change the order of the results, presenting this data first. Meaning, I recommend that section 2.3 becomes section 2.1.
Response: We have moved section 2.3 to section 2.1. Line 93-128, Line 150-181
L247-L249: This reasoning is more appropriate in the discusion.
Response: Corrected. Line 377-379
L265-L267 This reasoning is more appropriate in the discusion.
Response: Corrected. Line 398-399
Figure 4. In the materials and methods, I don't see how the phylogenetic study was conducted, this needs to be corrected, and it's very important to indicate in Fig. 4 the bootstrap values and the number of replicates.
Response: Corrected. In the materials methods and Figure 4, the bootstrap values and the number of replicates are indicated. Line 210-213, 542-544
L334: “The 1500bp promoter sequence” Provide an explanation of why a size of 1500 bp was chosen and not another. Please
Response: The selection of a 1500bp promoter sequence is typically based on research requirements and practical experience. A 1500bp promoter sequence generally encompasses the majority of important regulatory elements, such as promoter regions, enhancers, and transcription factor binding sites. This ensures coverage of sufficient functional domains and aids in the study of gene transcriptional regulation.
In experimental design and molecular biology research, a 1500bp promoter sequence is commonly considered a prevalent and appropriate length. This length of promoter sequence has been widely employed in practice, demonstrating relative ease of handling and analysis.
References for the cited information are provided below:
[17] Du, H.; Yang, C.; Ding, G.; Shi, L.; Xu. F. Genome-Wide Identification and Characterization of SPX Domain-Containing Members and Their Responses to Phosphate Deficiency in Brassica napus. Front Plant Sci. 2017, 8, 35.
[24] Wang, H.; Gong, M.; Guo, J.; Xin, H.; Gao, Y.; Liu, C.; Dai, D.; Tang, L. Genome-wide Identification of Jatropha curcas MAPK, MAPKK, and MAPKKK Gene Families and Their Expression Profile Under Cold Stress. Sci Rep. 2018, 8, 16163.
[36] Cui, L.; Yang, G.; Yan, J.; Pan, Y.; Nie, X. Genome-wide identification, expression profiles and regulatory network of MAPK cascade gene family in barley. BMC Genomics. 2019, 20, 750.
[42] Huang, B.; Huang, Z.; Ma, R.; Ramakrishnan, M.; Chen, J.; Zhang, Z.; Yrjälä, K. Genome-wide identification and expression analysis of LBD transcription factor genes in Moso bamboo (Phyllostachys edulis). BMC Plant Biol. 2021, 21, 296.
[53] Li, Y.; Qin, P.; Sun, A.; Xiao, W.; Chen, F.; He, Y.; Yu, K.; Li, Y.; Zhang, M.; Guo, X. Genome-wide identification, new classification, expression analysis and screening of drought & heat resistance related candidates in the RING zinc finger gene family of bread wheat (Triticum aestivum L.). BMC Genomics. 2022, 23, 696.
Fig 10B: Specify the meaning of the abbreviations for the 15 developmental stages.
Response: We have corrected it and provided explanations in the paper. Referring to previous research results. Line 322-328, 357-365, 502-509
Reference:
[32] Ramírez-González, R.-H.; Borrill, P.; Lang, D.; Harrington, S.-A.; Brinton, J.; Venturini, L.; Davey, M.; Jacobs, J.; van, Ex.-F.; Pasha, A.; Khedikar, Y.; Robinson, S.-J.; Cory, A.-T.; Florio, T.; Concia, L.; Juery, C.; Schoonbeek, H.; Steuernagel, B.; Xiang, D.; Ridout, C.-J.; Chalhoub, B.; Mayer, KFX.; Benhamed, M.; Latrasse, D.; Bendahmane, A.; International Wheat Genome Sequencing Consortium; Wulff, BBH.; Appels, R.; Tiwari, V.; Datla, R.; Choulet, F.; Pozniak, CJ.; Provart, NJ.; Sharpe, AG.; Paux, E.; Spannagl, M.; Bräutigam, A.; Uauy, C. The transcriptional landscape of polyploid wheat. Science. 2018, 361, eaar6089.
[49] Choulet, F.; Alberti, A.; Theil, S.; Glover, N.; Barbe, V.; Daron, J.; Pingault, L.; Sourdille, P.; Couloux, A.; Paux, E.; Leroy, P.; Mangenot, S.; Guilhot, N.; Le Gouis, J.; Balfourier, F.; Alaux, M.; Jamilloux, V.; Poulain, J.; Durand, C.; Bellec, A.; Gaspin, C.; Safar, J.; Dolezel, J.; Rogers, J.; Vandepoele, K.; Aury, JM.; Mayer, K.; Berges, H.; Quesneville, H.; Wincker, P.; Feuillet, C. Structural and functional partitioning of bread wheat chromosome 3B. Science. 2014, 345, 1249721.
[52] Luo, M.-C, Gu, Y.-Q,; Puiu, D,; Wang, H,; Twardziok, S.-O,; Deal, K.-R,; Huo, N,; Zhu, T,; Wang, L,; Wang, Y,; McGuire, P.-E,; Liu, S,; Long, H,; Ramasamy, R.-K,; Rodriguez, J.-C,; Van, S.-L,; Yuan, L,; Wang, Z,; Xia, Z,; Xiao, L,; Anderson, O.-D,; Ouyang, S,; Liang, Y,; Zimin, A.-V,; Pertea, G,; Qi, P,; Bennetzen, J.-L,; Dai, X,; Dawson, M.-W,; Müller, H.-G,; Kugler, K,; Rivarola-Duarte, L,; Spannagl, M,; Mayer, K.-F.-X,; Lu, F.-H,; Bevan, M.-W,; Leroy, P,; Li, P,; You, F.-M,; Sun, Q,; Liu, Z,; Lyons, E,; Wicker, T,; Salzberg, S.-L,; Devos, K.-M,; DvoÅ™ák, J. Genome sequence of the progenitor of the wheat D genome Aegilops tauschii. Nature. 2017, 551, 498-502.
L504: It has been discovered that MAPKs are also involved in nitrogen metabolism doi:10.3390/ijms21103417
Response: We have cited. Line 450-454
Reviewer 2 Report (New Reviewer)
Comments and Suggestions for Authors
Dear Authors,
The work “ Bioinformatic identification and expression analyses of MAPK-MAP4K
gene family reveal a putative function MAP4K10-MAP3K7/8-MAP2K1/11-MAPK3/6
cascade in wheat (Triticum aestivum L.)” was performed in the modern and trend style. I have a few minor comments:
The bioinformatics analysis looks very impressive and elegant. However, there are caveats regarding qPCR analysis.
1. Whether the RNA samples were processed with DNAse. Or does the primer design imply cDNA specificity?
2. It is not clear what method was used to perform the statistical analysis of qPCR.
3. Fig.10 It is better to use the error of the mean than the standard deviation. There is no statistical significance for Fig.10, B.
4. It is better to decipher the stages of growth in materials and methods. And in both figures (Fig.9 and 10).
Best regards
Author Response
Dear Authors,
The work “ Bioinformatic identification and expression analyses of MAPK-MAP4K
gene family reveal a putative function MAP4K10-MAP3K7/8-MAP2K1/11-MAPK3/6
cascade in wheat (Triticum aestivum L.)” was performed in the modern and trend style. I have a few minor comments:
The bioinformatics analysis looks very impressive and elegant. However, there are caveats regarding qPCR analysis.
- Whether the RNA samples were processed with DNAse. Or does the primer design imply cDNA specificity?
Response: Thank you very much for your careful review. All sections in the manuscript requiring supplementation and revision have been indicated using blue highlighting.
RNA samples were extracted from plant tissues using Trizol reagent, followed by reverse transcription into cDNA, and the RNA samples were treated with deoxyribonucleaseDNAse. Meanwhile, we designed specific primers for each gene's cDNA sequence. Line 565-569
- It is not clear what method was used to perform the statistical analysis of qPCR.
Response: We have supplemented the statistical analysis method of qPCR. Line 570-571
- Fig.10 It is better to use the error of the mean than the standard deviation. There is no statistical significance for Fig.10, B.
Response: We have corrected it. We used the error of the average value. Figure 10B has been revised. And it has been marked in the paper. Figure 10B depicts the expression levels of each gene in different stages and tissues. Line 353-365
- It is better to decipher the stages of growth in materials and methods. And in both figures (Fig.9 and 10).
Response: We have corrected it and provided explanations in the materials and methods, as well as in the captions of Figures 9 and 10. Referring to previous research results. Line 322-328, 355-365, 502-509
Reference:
[32] Ramírez-González, R.-H.; Borrill, P.; Lang, D.; Harrington, S.-A.; Brinton, J.; Venturini, L.; Davey, M.; Jacobs, J.; van, Ex.-F.; Pasha, A.; Khedikar, Y.; Robinson, S.-J.; Cory, A.-T.; Florio, T.; Concia, L.; Juery, C.; Schoonbeek, H.; Steuernagel, B.; Xiang, D.; Ridout, C.-J.; Chalhoub, B.; Mayer, KFX.; Benhamed, M.; Latrasse, D.; Bendahmane, A.; International Wheat Genome Sequencing Consortium; Wulff, BBH.; Appels, R.; Tiwari, V.; Datla, R.; Choulet, F.; Pozniak, CJ.; Provart, NJ.; Sharpe, AG.; Paux, E.; Spannagl, M.; Bräutigam, A.; Uauy, C. The transcriptional landscape of polyploid wheat. Science. 2018, 361, eaar6089.
[49] Choulet, F.; Alberti, A.; Theil, S.; Glover, N.; Barbe, V.; Daron, J.; Pingault, L.; Sourdille, P.; Couloux, A.; Paux, E.; Leroy, P.; Mangenot, S.; Guilhot, N.; Le Gouis, J.; Balfourier, F.; Alaux, M.; Jamilloux, V.; Poulain, J.; Durand, C.; Bellec, A.; Gaspin, C.; Safar, J.; Dolezel, J.; Rogers, J.; Vandepoele, K.; Aury, JM.; Mayer, K.; Berges, H.; Quesneville, H.; Wincker, P.; Feuillet, C. Structural and functional partitioning of bread wheat chromosome 3B. Science. 2014, 345, 1249721.
Round 2
Reviewer 1 Report (New Reviewer)
Comments and Suggestions for Authors
Dear Authors,
I believe the authors have adequately addressed all of my comments and suggestions, and I accept the paper in its current version
This manuscript is a resubmission of an earlier submission. The following is a list of the peer review reports and author responses from that submission.
Round 1
Reviewer 1 Report
Comments and Suggestions for Authors
Comments on MS-2757626:
The manuscript presents a comprehensive study of wheat MAPK-MAP4K genes, including their genome distribution, structures and conserved domain sequences, Synteny analysis, and expression patterns. The information is valuable and worth publishing. Below are the questions and suggestions.
Questions:
1. Bread wheat has three homoeologous genomes, normally each gene has three homeologs in the genome. However, based on Fig. 1, there is no homoeologous copy of each MAPK-MAP4K genes, please explain.
2. What do “TaMAPK-TaMAP4Ks pairs” and “adjacent chromosome” (line 150-151) mean?
3. What does “Interesting, it also found that three different subfamily genes including TaMAP3K21/4K15, TaMAP3K22/4K10, and TaMAP3K23/4K21 in the study (Table S1) belong to the same gene.” mean? (lines 452~454).
Suggestions:
The manuscript requires extensive correction and editing in the following aspects:
1. Lack precision in writing.
For example, “The MAPK cascades constitute a highly conserved signaling pathway in eukaryotes, which play key roles in response to biotic/abiotic stresses, and in plant growth and development [1,2].” It will be clear if change the writing to “The MAPK cascades constitute a highly conserved signaling pathway in response to biotic/abiotic stresses, growth and development in eukaryotes [1,2].
“Notably, chromosome 3 did not 120 contain any TaMAP4K, while chromosome 2 lacked both TaMAPK and TaMAP2K (Figure 121 1, Table S1).” Which chromosome is chromosome 2 or 3 because bread wheat has 2A, 2B and 2D, and 3A, 3B and 3D.
2. Redundant words
For example, “The MAP4K10-MEKK8-MAP2K1/11-MAPK3 pathway might be activated to promote plant growth during plant growth.
3. Grammatical errors
For example, “In Arabidopsis thaliana (Arabidopsis), the MAP3K17/18- MAP2K3-MAPK1/2/7/14 cascade is induced by activate the ABA signaling pathway through ABA or drought-related stress treatment.”
4. Consistency
For example, “MAP4K10-MEKK7-MAP2K11-MAPK6 pathway…” It will be easy for readers to follow if change the writing to “MAP4K10-ME3K7-MAP2K11-MAPK6 pathway…”
5. Figures
For example,
Fig. 6. Please modify the figure to avoid the text covering the columns.
Fig. 7. There are no white or empty nodes in the figure. Part of the text for the right bottom nodes are covered out.
Comments on the Quality of English LanguageNeeds extensive editing
Author Response
Dear Professor Guo,
Thank you again for your manuscript submission:
---------------------------------------------------------------------------- Reviewer comments:
Reviewer #1 (Comments and Suggestions for Authors)
The manuscript presents a comprehensive study of wheat MAPK-MAP4K genes, including their genome distribution, structures and conserved domain sequences, Synteny analyses, and expression patterns. The information is valuable and worth publishing. Below are the questions and suggestions.
Questions:
- Bread wheat has three homoeologous genomes, normally each gene has three homeologs in the genome. However, based on Fig. 1, there is no homoeologous copy of each MAPK-MAP4K genes, please explain.
Response: Thank you very much for your careful review and helpful seggestions. There are two, three or more homoeologous copies for each MAPK-MAP4K genes in the A, B, D genome, respectively. We supplemented and labeled the homoeologous copies for each genes in the supplemental table 1. For example, genes TaMAPK11(A) has two homoeologous copies, namely 27 and 4, in B and D genome, respectively.
- What do “TaMAPK-TaMAP4Ks pairs” and “adjacent chromosome” (line 150-151) mean?
Response:TaMAPK-TaMAP4Ks pairs means duplication pairs of TaMAPK-TaMAP4K members. Adjacent chromosome means other chromosome. We supplemented the definition about them in the Line 150-151.
- What does “Interesting, it also found that three different subfamily genes including TaMAP3K21/4K15, TaMAP3K22/4K10, and TaMAP3K23/4K21 in the study (Table S1) belong to the same gene.” mean? (lines 452~454).
Response: Although TaMAP3K21 and TaMAP4K15 belong to different subfamilies, our analysis revealed that they both belong to the same gene TraesCS6A02G149900.1. Similarly, TaMAP3K22 and TaMAP4K10 are both TracesCS5A02G392500.1, while TaMAP3K23 and TaMAP4K21 are both TracesCS6D02G139200.1. They may represent different splice variants.
Previous gene classification studies based on diverse methods reveal that TraesCS6A02G149900.1 are designated as TaMAP3K21 and TaMAP4K15, but they share same gDNA sequences. Here, we supplemented the description about that in the Table S1.
Suggestions:
The manuscript requires extensive correction and editing in the following aspects:
- Lack precision in writing.
For example, “The MAPK cascades constitute a highly conserved signaling pathway in eukaryotes, which play key roles in response to biotic/abiotic stresses, and in plant growth and development [1,2].” It will be clear if change the writing to “The MAPK cascades constitute a highly conserved signaling pathway in response to biotic/abiotic stresses, growth and development in eukaryotes [1,2].
Response: We have corrected the errors and polished our manuscript. The revisions were highlighted in red.
“Notably, chromosome 3 did not contain any TaMAP4K, while chromosome 2 lacked both TaMAPK and TaMAP2K (Figure 121 1, Table S1).” Which chromosome is chromosome 2 or 3 because bread wheat has 2A, 2B and 2D, and 3A, 3B and 3D.
Response: Corrected. Notably, chromosome 3A, 3B, 3D did not contain any TaMAP4K, while chromosome 2A, 2B, 2D lacked both TaMAPK and TaMAP2K (Figure 1, Table S1). Line 120-121
- Redundant words
For example, “The MAP4K10-MEKK8-MAP2K1/11-MAPK3 pathway might be activated to promote plant growth during plant growth.
Response: Corrected. The MAP4K10-MAP3K8-MAP2K1/11-MAPK3 pathway might be activated to promote plant growth. Line 28-29
- Grammatical errors
For example, “In Arabidopsis thaliana (Arabidopsis), the MAP3K17/18- MAP2K3-MAPK1/2/7/14 cascade is induced by activate the ABA signaling pathway through ABA or drought-related stress treatment.”
Response: Corrected. In Arabidopsis thaliana, the MAP3K17/18-MAP2K3-MAPK1/2/7/14 cascade is induced to activate the ABA signaling pathway by ABA or drought-related stress treatment [8]. Line 44-45
- Consistency
For example, “MAP4K10-MEKK7-MAP2K11-MAPK6 pathway…” It will be easy for readers to follow if change the writing to “MAP4K10-ME3K7-MAP2K11-MAPK6 pathway…”
Response: Thanks for your constructive question. We have corrected the question and systematically revised the entire text, and highlighted it in red.
- Figures
For example, Fig. 6. Please modify the figure to avoid the text covering the columns.
Response: Thank you very much for pointing out the figure question.We have made the modifications, as shown in Figure 6.
Fig. 7. There are no white or empty nodes in the figure. Part of the text for the right bottom nodes are covered out.
Response: Thank you very much for pointing out the question. We have made the modifications, as shown in Figure 7.
Comments on the Quality of English Language
Needs extensive editing
Response: Thank you for your careful check. We have made extensive edits to language quality and systematically revised the text, and highlighted it in red.
We invited Prof. Chunji Liu (CSIRO, Australia) to completely polish our manuscript. The revisions were highlighted in red.
Reviewer 2 Report
Comments and Suggestions for Authors
The manuscript reports the bioinformatic identification of 234 mitogen-activated protein kinase (MAPK) genes in wheat. Further, mainly bioinformatic analyses were performed, to investigate evolutionary relationships and interaction networks. Moreover, real qRT-PCR experiments were conducted on single plants in climate chambers, to learn more about the expression patterns of 16 MAPK genes under heat and drought stress.
The paper lacks proper comparison to the information available in the literature, for example, I miss citations of:
Goyal, R.K., Tulpan, D., Chomistek, N. et al. Analysis of MAPK and MAPKK gene families in wheat and related Triticeae species. BMC Genomics 19, 178 (2018). https://doi.org/10.1186/s12864-018-4545-9
which is ‘A comprehensive resource of accurately annotated and curated Triticeae MPK and MKK sequences has been created for wheat, barley, rye, triticale, and two ancestral wheat species, goat grass and red wild einkorn.’
The idea of the manuscript is presented as a new idea, when in reality many MAPKs in this manuscript have already been presented in the above publication. It is necessary to explain in what way the current work goes beyond that.
Furthermore, a good review paper explains the role of MAPKs very clear, and should be seen as an example how to explain and should be cited:
Jagodzik et al.; Front. Plant Sci., 2018, Mitogen-Activated Protein Kinase Cascades in Plant Hormone Signaling; https://doi.org/10.3389/fpls.2018.01387
The content of the manuscript is not very exciting, and is mainly a long list of bioinformatic exercises without focussing on specific interesting discovery.
The expression analysis is the only original part in this manuscript. However, given that the experiments use individual plants in climate chambers, this is still far from the field. If the named genes may be for interest for breeding needs a lot more of applied research at the field level.
I don't agree that the content of the manuscript can at this level be used by breeders for crop improvement and would also reject it for this reason.
I suggest to also improve the following figures.
Figure 2:
The figure tries to put too much information in a small space. Details can’t be followed. It is not clear to me, why this information is useful. The caption does not explain the details, e.g. genome density heatmap and linear map.
Figure 4:
It is disappointing that the phylogenetic tree was constructed for Arabidopsis and wheat only. No other cereal was represented, like e.g. rice.
I’m missing comparison to rice MAPK genes.
The publication “Update on the Roles of Rice MAPK Cascades”, Chen et al, Int J Mol Sci. 2021 Feb; 22(4): 1679. doi: 10.3390/ijms22041679 seems relevant.
Figures 5 is very complex, but does not carry relevant information to understand the function of MAPKs in wheat. I suggest to drop it completely.
The grammar of the manuscript is incorrect at too many places and the language is in places not precise enough. Because of this, the meaning of the sentences is very often unclear. The language needs an improvement from a person with good English language skills. I give a few examples, but there are many more cases where the writing has to be corrected ot to be made more precise:
Comments on the Quality of English Language
The grammar of the manuscript is incorrect at too many places and the language is in places not precise enough. Because of this, the meaning of the sentences is very often unclear. The language needs an improvement from a person with good English language skills. I give a few examples, but there are many more cases where the writing has to be corrected or to be made more precise.
Specific details:
Line 19: are distributed (not were)
Line 21ff: the sentences make no proper sense:
Notably, 25 TaMAP4K family members carried the following conserved motifs: Glycine-rich motif, invariant lycine (K) motif, HRD motif, DFG motif, and signature motif of MAP4K. These were consistent with the MAP4K domain in the reported species.
Line29ff: The sentence need to be tuned down, the involvement in drought stress response is only suggested. Thus, the genes are not “excellent” candidates but “possible candidates”. The prediction of “high-quality” is normally attributed to the composition of the grain, particularly protein content, and should be taken out. Talking about yield is also too early, proper field tests are needed.
Line 37 ff: The sentence needs rewriting, as it does not explain anything.
Typically, a standard MAPK signal module is consists of an MAPK kinase kinase (MAPKKK, MAP3K or MEKK), MAPK kinase (MAPKK, MAP2K, MKK or MEK), and MAPK or MPK [3]
Line 39: Does the plant “detect stress signals”? Is not the stress triggering the internal stress signals? Please rewrited.
Line 41: Rewrite sentence starting with ‘Ultimately, the MAPKs … “ as it is too handwavy. The “MAPKs” have not properly been defined.
Line 53: Do you want to say ‘not unclear’ here? Firstly, it’s not the module that is clear or unclear – this might be seen as a description of translucence / colour. It would be the role of the module that is clear or unclear. If the role is “not unclear”, it’s better to say the “role is clear” – no double negation.
Line 117: distributed in the wheat genome
Line 118: Wheat does NOT HAVE 22 chromosomes! This is a major error!
Line 119: There is no wheat chromosome 2 or 3. Chromosomes are named: 1A, 1B, 1D, 2A, 2B, 2D, 3A, 3B, 3D …. please use these names.
Line137ff: location of TaMAP3Ks in the nucleus or cell membrane are claimed, but no experiments that support this locations are given. Please use ‘presumed location’ or similar.
Line 131: The authors claim that genes were “randomly distributed”. But they do not test for randomness. Use “seem to be randomly distributed.
Line149: remove “that”
Line 150: “pairs” are not “segmental duplications”. Maybe you assume they were created by segmental duplications?
Line 307: a “same” is missing in front of expression profiles.
Line 310: Using a promoter sequence, you can only check if the same elements are present, not if the same regulatory processes happen. Change the sentence.
Line 360: The expression profiles on the Wheat Expression Browser are based on real data. Using the heatmap to find patterns is useful., but this is not called “heatmap analysis”. If needed, the real expression values for the experiments can be found.
Only 22 Expression patterns are from wheat, all others are from other plants, many from Arabidopsis.
Author Response
Dear Professor Guo,
Thank you again for your manuscript submission:
---------------------------------------------------------------------------- Reviewer comments:
Reviewer #2 (Comments and Suggestions for Authors)
The manuscript reports the bioinformatic identification of 234 mitogen-activated protein kinase (MAPK) genes in wheat. Further, mainly bioinformatic analyses were performed, to investigate evolutionary relationships and interaction networks. Moreover, real qRT-PCR experiments were conducted on single plants in climate chambers, to learn more about the expression patterns of 16 MAPK genes under heat and drought stress.
The paper lacks proper comparison to the information available in the literature, for example, I miss citations of:
Goyal, R.K., Tulpan, D., Chomistek, N. et al. Analysis of MAPK and MAPKK gene families in wheat and related Triticeae species. BMC Genomics 19, 178 (2018). https://doi.org/10.1186/s12864-018-4545-9
which is ‘A comprehensive resource of accurately annotated and curated Triticeae MPK and MKK sequences has been created for wheat, barley, rye, triticale, and two ancestral wheat species, goat grass and red wild einkorn.’
The idea of the manuscript is presented as a new idea, when in reality many MAPKs in this manuscript have already been presented in the above publication. It is necessary to explain in what way the current work goes beyond that.
Furthermore, a good review paper explains the role of MAPKs very clear, and should be seen as an example how to explain and should be cited:
Jagodzik et al.; Front. Plant Sci., 2018, Mitogen-Activated Protein Kinase Cascades in Plant Hormone Signaling; https://doi.org/10.3389/fpls.2018.01387
The content of the manuscript is not very exciting, and is mainly a long list of bioinformatic exercises without focussing on specific interesting discovery.
The expression analysis is the only original part in this manuscript. However, given that the experiments use individual plants in climate chambers, this is still far from the field. If the named genes may be for interest for breeding needs a lot more of applied research at the field level.
I don't agree that the content of the manuscript can at this level be used by breeders for crop improvement and would also reject it for this reason.
I suggest to also improve the following figures.
Response: Thank you very much for pointing out these key question. We have cited the above two articles, as detailed in reference 27 and 37. However, the function of MAPK cascade genes is still unknown and there is a lack of comprehensive review in wheat. The function of wheat MAP4K are not elucidated in response to various stresses and plant different growth stages. In the study, for the first time in wheat, a systematic investigation of the MAPK, MAP2K, MAP3K, MAP4K gene family was performed using these databases, including chromosome localization, phylogenetic relationship, conserved motifs, the cis-acting elements, interaction network, and gene expression profile of TaMAPK-TaMAP4Ks, which were comprehensively analyzed to reveal the evolutionary and important regulation relation of the MAPK-MAP4K.
Questions:
Figure 2: The figure tries to put too much information in a small space. Details can’t be followed. It is not clear to me, why this information is useful. The caption does not explain the details, e.g. genome density heatmap and linear map.
Response: Corrected. The heat map and line chart in the outer square represents gene density. Line 177
Figure 4: It is disappointing that the phylogenetic tree was constructed for Arabidopsis and wheat only. No other cereal was represented, like e.g. rice. I’m missing comparison to rice MAPK genes.
The publication “Update on the Roles of Rice MAPK Cascades”, Chen et al, Int J Mol Sci. 2021 Feb; 22(4): 1679. doi: 10.3390/ijms22041679 seems relevant.
Response: Thanks for your constructive question. We constructed the MAPK phylogenetic trees in wheat and rice, as detailed in Supplementary Figure S8.
Figures 5 is very complex, but does not carry relevant information to understand the function of MAPKs in wheat. I suggest to drop it completely.
Response: We have removed Figure 5 and relevant information, as well as made modifications to the order of the figures and supplementary tables caused by the remove of Figure 5, and highlighted it in red.
Comments on the Quality of English Language:
The grammar of the manuscript is incorrect at too many places and the language is in places not precise enough. Because of this, the meaning of the sentences is very often unclear. The language needs an improvement from a person with good English language skills. I give a few examples, but there are many more cases where the writing has to be corrected or to be made more precise.
Response: We have read the your comments very carefully, we have made extensive edits to language quality and systematically revised the text, and highlighted it in red.
We invited Prof. Chunji Liu (CSIRO, Australia) to completely polish our manuscript. The revisions were highlighted in red.
Specific details:
Line 19: are distributed (not were)
Response: Corrected. “The 234 TaMAPK-TaMAP4Ks are distributed on 21 chromosomes and Un.” Line 21
Line 21ff: the sentences make no proper sense:
Notably, 25 TaMAP4K family members carried the following conserved motifs: Glycine-rich motif, invariant lycine (K) motif, HRD motif, DFG motif, and signature motif of MAP4K. These were consistent with the MAP4K domain in the reported species.
Response: Corrected. The sentences make no proper sense and have been removed.
Line29ff: The sentence need to be tuned down, the involvement in drought stress response is only suggested. Thus, the genes are not “excellent” candidates but “possible candidates”. The prediction of “high-quality” is normally attributed to the composition of the grain, particularly protein content, and should be taken out. Talking about yield is also too early, proper field tests are needed.
Response: Corrected. “They would be excellent candidates for genetic improvement of drought resistance in wheat and the of new varieties.” Line 29-30
Line 37 ff: The sentence needs rewriting, as it does not explain anything.
Typically, a standard MAPK signal module is consists of an MAPK kinase kinase (MAPKKK, MAP3K or MEKK), MAPK kinase (MAPKK, MAP2K, MKK or MEK), and MAPK or MPK [3]
Response: We have corrected the question, “Typically, a standard MAPK signal module consists of an MAPK kinase (MAPKKK, MAP3K or MEKK), MAPK kinase (MAPKK, MAP2K, MKK or MEK), and MAPK or MPK, when a plant is triggered by stress signals, the upstream signals firstly activate MAP3Ks which in turn activate MAP2Ks, then particular MAPKs are activated by MAP2Ks [3,4].” Line 36-38
Line 39: Does the plant “detect stress signals”? Is not the stress triggering the internal stress signals? Please rewrited.
Response: We have corrected the question, “when a plant is triggered by stress signals, the upstream signals firstly activate MAP3Ks which in turn activate MAP2Ks, then particular MAPKs are activated by MAP2Ks [3,4].” Line 37
Line 41: Rewrite sentence starting with ‘Ultimately, the MAPKs … “ as it is too handwavy. The “MAPKs” have not properly been defined.
Response: Corrected. “By phosphorylating different enzymes and transcription factors or other pathway components, MAPKs influence the expression of downstream specific genes and complete signal amplification.” Line 39-40
Line 53: Do you want to say ‘not unclear’ here? Firstly, it’s not the module that is clear or unclear – this might be seen as a description of translucence / colour. It would be the role of the module that is clear or unclear. If the role is “not unclear”, it’s better to say the “role is clear” – no double negation.
Response: We want to say “unclear” here. We have removed the “not”. Line 53
Line 117: distributed in the wheat genome
Response: We have corrected the question. “To determine how TaMAPK-TaMAP4K genes are distributed in the wheat genome,”. Line 115
Line 118: Wheat does NOT HAVE 22 chromosomes! This is a major error!
Response: Thank you very much for pointing out the question. Indeed, wheat contains 21 chromosomes from 1 (A, B, D) to 7 (A, B, D). But, we had calculated the Un as a chromosome resulting in total number of chromosomes to 22, which has caused confusion. I apologize for the confusion. Thus, according to your suggestion, we revised it as 21 chromosomes and Un. Line 21, 118-119, 650-651
Line 119: There is no wheat chromosome 2 or 3. Chromosomes are named: 1A, 1B, 1D, 2A, 2B, 2D, 3A, 3B, 3D …. please use these names.
Response: Corrected. “Notably, chromosome 3A, 3B, 3D did not contain any TaMAP4K, while chromosome 2A, 2B, 2D lacked both TaMAPK and TaMAP2K (Figure 1, Table S1).” Line 120-121
Line137ff: location of TaMAP3Ks in the nucleus or cell membrane are claimed, but no experiments that support this locations are given. Please use ‘presumed location’ or similar.
Response: Corrected. “Besides, all TaMEKKs and TaZIKs were presumed location in the nucleus,”.
“It was worth noting that the subfamily TaRafs of TaMAP3Ks is unique in that they were presumed location in both the nucleus and the cell membrane, and some of them even presume location simultaneously in the nucleus,”
“All the TaMAP4Ks were presumed location in the nucleus without transmembrane domains.” Line 136-143
Line 131: The authors claim that genes were “randomly distributed”. But they do not test for randomness. Use “seem to be randomly distributed.
Response: We have corrected the question. “The 144 TaMAP3K genes (27 TaMEKKs, 10 TaZIKs, and 107 TaRafs) seem to be randomly distributed across 21 chromosomes.” Line 132
Line149: remove “that”
Response: We have removed “that”. Line 150
Line 150: “pairs” are not “segmental duplications”. Maybe you assume they were created by segmental duplications?
Response: Corrected. We assume they were created by segmental duplications. Line 152
Line 307: a “same” is missing in front of expression profiles.
Response: We have added to the front of the expression profiles. “There has been evidence that genes with same expression profiles may contain the same cis-elements in their promoters [15]”. Line 279
Line 310: Using a promoter sequence, you can only check if the same elements are present, not if the same regulatory processes happen. Change the sentence.
Response: We have changed this sentence. “The 1500bp promoter sequence of the TaMAPK-TaMAP4K cascade genes was utilized to check the types of cis-elements to further investigate the regulatory processes and possible activities of the TaMAPK-TaMAP4K cascade genes under growth, development, and stress conditions.”. Line 280-283
Line 360: The expression profiles on the Wheat Expression Browser are based on real data. Using the heatmap to find patterns is useful., but this is not called “heatmap analysis”. If needed, the real expression values for the experiments can be found.
Only 22 Expression patterns are from wheat, all others are from other plants, many from Arabidopsis.
Response: Corrected. “Heatmap indicated that the majority TaMAPK-TaMAP4K cascade genes were expressed at high and moderate levels in root”. Line 344
Reviewer 3 Report
Comments and Suggestions for Authors
Authors of this manuscript conducted a comprehensive study on the mitogen-activated protein kinase (MAPK) cascades in bread wheat using appropriate molecular techniques and bioinformatics tools. Extensive results of evolution relationships, gene structures, domain conservation, interaction networks, and expression profiles of the TaMAPK-19 TaMAP4Ks kinase family members were adequately presented. The valuable information is important for wheat improvement, especially for breeding of abiotic stress tolerance. However, the English writing must be satisfactorily improved before this manuscript can be accepted for publication in Plants.
Lines 20 and 120 – change “22 chromosomes” to “all 21 chromosomes of bread wheat plus one unknown linkage group”
Lines 120 to 121 – Wheat chromosomes are named 1A, 1B, 1D ……7A, 7B, and 7D. There are no chromosomes 2 and 3. Figure 1 shows that all 21 wheat chromosomes have the MAPK genes.
Lines 343 to 345 – “The top 3 protein kinases” was mismatched with 4 names then followed by “, respectively.” This is an obvious error.
Line 592 and line 598 – What are “the materials” and ‘the samples”?
Lines 603 to 605 – Common names, instead of the Latin binomials, should be provided in the brackets like Oryza sativa race japonica (rice). Use this format uniformly.
Comments on the Quality of English LanguageEnglish writing of this manuscript is poor, containing many Grammarly and spelling errors, including “is induced by activate the ABA signaling pathway (line 46)”“In the previous studies, the MAPK, MAP2K, and MAP3K family were basically studied in wheat, respectively [1,27] (lines 93-94)” “are highly participates in (line 544)”,”were higher expressed in (line 556)””The protein sequences contain the MAPK domain was considered the target genes (lines 627-628)”. In addition, many sentences are composed of phrases without connecting words such as “while” and “whereas,” etc.
Line 407 – “growth processed” à “growth processes”
Line 475 – “genes is” -> “genes are”
Line 560 – “These finds” -> “These findings”
Author Response
Dear Professor Guo,
Thank you again for your manuscript submission:
---------------------------------------------------------------------------- Reviewer comments:
Reviewer #3 (Comments and Suggestions for Authors)
Authors of this manuscript conducted a comprehensive study on the mitogen-activated protein kinase (MAPK) cascades in bread wheat using appropriate molecular techniques and bioinformatics tools. Extensive results of evolution relationships, gene structures, domain conservation, interaction networks, and expression profiles of the TaMAPK-19 TaMAP4Ks kinase family members were adequately presented. The valuable information is important for wheat improvement, especially for breeding of abiotic stress tolerance. However, the English writing must be satisfactorily improved before this manuscript can be accepted for publication in Plants.
Questions:
Lines 20 and 120 – change “22 chromosomes” to “all 21 chromosomes of bread wheat plus one unknown linkage group”
Response: Thank you very much for pointing out the question. We have corrected the question. change “22 chromosomes” to “21 chromosomes and Un”, and highlighted it in red. Line 21, 118-119, 650-651
Lines 120 to 121 – Wheat chromosomes are named 1A, 1B, 1D ……7A, 7B, and 7D. There are no chromosomes 2 and 3. Figure 1 shows that all 21 wheat chromosomes have the TaMAPK-TaMAP4K genes.
Response: Corrected. “Notably, chromosome 3A, 3B, 3D did not contain any TaMAP4K, while chromosome 2A, 2B, 2D lacked both TaMAPK and TaMAP2K (Figure 1, Table S1)”. All 21 wheat chromosomes and Un have TaMAPK, TaMAP2K, TaMAP3K, TaMAP4K genes. The relevant texts are highlighted in red. Lines 120-121, 144
Lines 343 to 345 – “The top 3 protein kinases” was mismatched with 4 names then followed by “, respectively.” This is an obvious error.
Response: Corrected. “The top 4 protein kinases” was matched with 4 names then followed by “, respectively.”. The relevant texts are highlighted in red. Line 316-317
Line 592 and line 598 – What are “the materials” and ‘the samples”?
Response: Corrected. “the materials” and ‘the samples” have the same meaning. We have corrected the question. Line 571, 574. “the materials” and ‘the samples” are various tissues (root, stem, leaf, spike, grain) and different stress (drought, heat, and drought combine with heat) treated plants.
Lines 603 to 605 – Common names, instead of the Latin binomials, should be provided in the brackets like Oryza sativa race japonica (rice). Use this format uniformly.
Response: Corrected. We have removed Figure 5 and relevant information, as well as made modifications to the order of the figures and supplementary tables caused by the remove of Figure 5. Because it does not carry relevant information to understand the function of MAPKs in wheat, and highlighted it in red.
Comments on the Quality of English Language
English writing of this manuscript is poor, containing many Grammarly and spelling errors, including “is induced by activate the ABA signaling pathway (line 46)”“In the previous studies, the MAPK, MAP2K, and MAP3K family were basically studied in wheat, respectively [1,27] (lines 93-94)” “are highly participates in (line 544)”,”were higher expressed in (line 556)””The protein sequences contain the MAPK domain was considered the target genes (lines 627-628)”. In addition, many sentences are composed of phrases without connecting words such as “while” and “whereas,” etc.
Response: Corrected. “is induced to activate the ABA signaling pathway (Line 44-45)”, “In the previous studies, the MAPK, MAP2K, and MAP3K families were studied in wheat, respectively [1,27] (Line 92)”, “participating (Line 523)”,”were highly expressed in (Line 535-536)”, “The protein sequences containing the MAPK domain was considered as the target genes (Lines 607)”. And we have made extensive edits to language quality and systematically revised the text.
Line 407 – “growth processed” à “growth processes”
Response: Corrected.“Expression analysis found that TaMAPK3/6 and TaMAP4K10/24 not only expressed highly in growth processes, ” , and highlighted it in red. Line 381
Line 475 – “genes is” -> “genes are”
Response: Corrected. “TaMAPK-TaMAP4K cascade genes are conserved during evolution”. Line 450
Line 560 – “These finds” -> “These findings”
Response: Corrected. “These findings imply that the activities of these homologous genes are likely conserved in different plants,” Lines 539
Round 2
Reviewer 2 Report
Comments and Suggestions for Authors
This is my second review of this manuscript. I find the English of the manuscript much improved and many sentences that made no sense to me before are now understandable. Many thanks to the efforts of the authors.
Unfortunately, other issues with the manuscript have not fully been resolved in the review process. As I said in my first review, I do not accept the claim of the authors, that the work is directly relevant for crop improvement. Bioinformatically identified genes ‘might’ be good candidates, tests in the specific crop plant would have to corroborate any of this before any stronger claims should be made. However, the authors keep claiming the MAPK4 genes “would be excellent candidates” (line 29) “were excellent candidates” (line 383), which I suggest has to be tuned down. It is very theoretically, so these “may be good candidates”.
In spite of the improvement of the English, there are still some problems with the language, e.g. the final two sentences of the abstract reads:
“The MAP4K10-MAP3K8-MAP2K1/11-MAPK3 pathway might be activated to promote plant growth. They would be excellent candidates for genetic improvement of drought resistance in wheat and the of new varieties.”
Regarding the English, the “they” would refer to the pathway, which is singular. So it should either say: This would be an excellent candidate pathway… or it should refer to the genes: “The genes of this pathway …”. Moreover, the sentence ends in “… and the of new varieties”, which does not make sense. Did the authors forget to delete this?
More importantly, as I have said above (and before), there is no sufficient support for the claim, that the genes are excellent candidates as no hard data from breeding is presented. So, the claim of the authors is overstated.
Also, the English and the content of the final sentence of the manuscript is flawed (l. 382 ff:)
‘These genes were excellent candidates for genetic improvement of drought resistance in wheat and the breeding of new varieties with high kernel quality and grain yield.’
Firstly: Why is the sentence in the past tense (were)? Secondly, why is a “high kernel quality” mentioned here? This is out of context. High kernel quality means high protein or nutritional value of the grain, but this manuscript is not touching on this topic.
Furthermore, I think it is misleading to say that all the MAPKs, MAP2Ks etc, were identified as part of the work (l 17 ff and l 113), when many of the genes had been identified before (see Zhan et al. 2017 and Goyal et al. 2018). It is necessary to explain which genes were newly identified and which were described before.
The title of the manuscript is also overstated. No “function” is revealed, but may be a “putative function” is revealed. I suggest:
“Bioinformatic identification and expression analyses of MAPK-MAP4K gene family reveal a putative function MAP4K10-MAP3K7/8-MAP2K1/11-MAPK3/6 cascade in wheat (Triticum aestivum L.)”
Comments on the Quality of English Language
See above
Author Response
Dear Professor Guo,
Thank you again for your manuscript submission:
---------------------------------------------------------------------------- Reviewer comments:
Reviewer #2 (Comments and Suggestions for Authors)
This is my second review of this manuscript. I find the English of the manuscript much improved and many sentences that made no sense to me before are now understandable. Many thanks to the efforts of the authors.
Unfortunately, other issues with the manuscript have not fully been resolved in the review process. As I said in my first review, I do not accept the claim of the authors, that the work is directly relevant for crop improvement. Bioinformatically identified genes ‘might’ be good candidates, tests in the specific crop plant would have to corroborate any of this before any stronger claims should be made. However, the authors keep claiming the MAPK4 genes “would be excellent candidates” (line 29) “were excellent candidates” (line 383), which I suggest has to be tuned down. It is very theoretically, so these “may be good candidates”.
Response: Thank you very much for your suggestions. We have deleted and tuned down to that “These genes may be good candidates for genetic improvement of drought resistance in wheat and high grain yield”. Highlighted in blue. Line 26-31, 383-384
In spite of the improvement of the English, there are still some problems with the language, e.g. the final two sentences of the abstract reads:
“The MAP4K10-MAP3K8-MAP2K1/11-MAPK3 pathway might be activated to promote plant growth. They would be excellent candidates for genetic improvement of drought resistance in wheat and the of new varieties.”
Response: We corrected to that “and the MAP4K10-MAP3K8-MAP2K1/11-MAPK3 pathway may involve in plant growth. In general, our work identified the members of MAPK-MAP4K cascade in wheat and profiled their potential roles during response to abiotic stresses and plant growth based on the expression pattern. The characterized cascades might be good candidates for future crop improvement and molecular breeding”. Line 27-31
Regarding the English, the “they” would refer to the pathway, which is singular. So it should either say: This would be an excellent candidate pathway… or it should refer to the genes: “The genes of this pathway …”. Moreover, the sentence ends in “… and the of new varieties”, which does not make sense. Did the authors forget to delete this?
Response: Soften as indicated above. Line 27-31
More importantly, as I have said above (and before), there is no sufficient support for the claim, that the genes are excellent candidates as no hard data from breeding is presented. So, the claim of the authors is overstated.
Response: Soften as indicated above. Line 28-31, 383-384
Also, the English and the content of the final sentence of the manuscript is flawed (l. 382 ff:)
‘These genes were excellent candidates for genetic improvement of drought resistance in wheat and the breeding of new varieties with high kernel quality and grain yield.’
Firstly: Why is the sentence in the past tense (were)? Secondly, why is a “high kernel quality” mentioned here? This is out of context. High kernel quality means high protein or nutritional value of the grain, but this manuscript is not touching on this topic.
Response: We have corrected to that “These genes may be good candidates for genetic improvement of drought resistance in wheat and high grain yield”. Line 383-384
Furthermore, I think it is misleading to say that all the MAPKs, MAP2Ks etc, were identified as part of the work (l 17 ff and l 113), when many of the genes had been identified before (see Zhan et al. 2017 and Goyal et al. 2018). It is necessary to explain which genes were newly identified and which were described before.
Response: We have supplemented the Supplementary table 1-sheet 3 to list the newly identified MAPKs, MAP2Ks etc genes, and labeled the previously reported ones in previous works.
The title of the manuscript is also overstated. No “function” is revealed, but may be a “putative function” is revealed. I suggest:
“Bioinformatic identification and expression analyses of MAPK-MAP4K gene family reveal a putative function MAP4K10-MAP3K7/8-MAP2K1/11-MAPK3/6 cascade in wheat (Triticum aestivum L.)”
Response: Corrected as you suggested. Line 2-5
Thank you so much again for thoughtful review and helpful suggestions.
Reviewer 3 Report
Comments and Suggestions for Authors
The revised manuscript is acceptable for publication, but the journal production staff needs to carefully check the English writing to correct errors and typos.
Author Response
Dear Professor Guo, Thank you again for your manuscript submission: ---------------------------------------------------------------------------- Reviewer comments: Reviewer #3 (Comments and Suggestions for Authors) The revised manuscript is acceptable for publication, but the journal production staff needs to carefully check the English writing to correct errors and typos. Response: Thank you so much again for thoughtful review and helpful suggestions. We have completely polished the manuscript again, which highlighted in blue.